# Direct comparisons of neural activity during placebo analgesia and nocebo hyperalgesia between humans and rats
Damien C. Boorman [1,2,3] ✉, Lewis S. Crawford[1,2], Luke A. Henderson[1,2] & Kevin A. Keay [1,2]

Placebo analgesia and nocebo hyperalgesia can profoundly alter pain perception, offering critical implications for pain management. While animal models are increasingly used to explore the underlying mechanisms of these phenomena, it remains unclear whether animals experience placebo and nocebo effects in a manner comparable to humans or whether the associated neurobiological pathways are conserved across species. In this study, we introduce a novel framework for comparing brain activity between humans and rodents during placebo analgesia and nocebo hyperalgesia. Using c-Fos immunohistochemistry in rats and fMRI in humans, we examined neural activity in 70 pain-related brain regions, identifying both conserved and species-specific connectivity changes. Functional connectivity analysis, refined by pruning connections based on known anatomical pathways, revealed significant overlap in key regions, including the amygdala, anterior cingulate cortex, and nucleus accumbens, highlighting conserved circuits driving placebo and nocebo responses. This cross-species methodology offers a powerful new approach for investigating the neurobiology of pain modulation, bridging the gap between animal models and human studies. Identifying these common connections validates the use of animal models and enables preclinical researchers to focus on circuits that are conserved across species, ensuring greater translational relevance when developing new and effective treatments for pain conditions.

Chronic pain affects millions of individuals worldwide, and current treatments often provide inadequate relief, leading to significant physical, emotional, and economic burdens. Recent research has highlighted the potential utility of placebo analgesia as a powerful, non-pharmacological intervention that could augment chronic pain management by harnessing the brain's endogenous pain modulation systems[1]. Conversely, nocebo hyperalgesia—the phenomenon in which negative expectations worsen pain—may significantly contribute to both the development and persistence of chronic pain conditions[2]. Identifying ways to inhibit the nocebo effect could therefore be transformative in chronic pain therapy, potentially preventing its escalation and improving patient outcomes.

Historically, studies on placebo analgesia and nocebo hyperalgesia have been primarily limited to humans, which have provided critical insights into the psychological and neurobiological mechanisms driving these effects. Only recently have researchers begun to develop and validate animal models to study these phenomena in more controlled experimental settings[3–7]. Rodent models enable the experimental manipulation of neural

circuits and molecular pathways that are not possible in human studies, allowing us to identify the cellular and synaptic mechanisms driving these pain modulatory processes. Moreover, rodent models provide an avenue for testing potential pharmacological or genetic interventions that could mitigate nocebo hyperalgesia or enhance placebo analgesia, which could lead to novel therapeutic strategies for pain conditions.

Significant progress has been made in mapping the neurobiological substrates of placebo analgesia and nocebo hyperalgesia. Studies in humans have implicated key brain regions, including the prefrontal cortex, anterior cingulate cortex (ACC), insular cortex and midbrain periaqueductal gray (PAG), which are involved in top-down modulation of pain through cognitive and emotional regulation[8,9]. Similarly, in rodent models, structures including the frontal cortical regions, amygdala, ACC, and PAG have also been shown to play central roles in modulating pain responses during placebo and nocebo conditions[10,11]. Furthermore, pathways involving the amygdala and nucleus accumbens (NAc), which regulate emotional processing and reward, have emerged as critical mediators in both placebo-

[1]School of Medical Sciences (Neuroscience), Faculty of Medicine and Health, The University of Sydney, Sydney, NSW, Australia. [2]Brain and Mind Centre, The University of Sydney, Sydney, NSW, Australia. [3]Department of Psychology, University of Toronto Mississauga, Mississauga, Ontario, Canada. ✉e-mail: dboo2217@uni.sydney.edu.au

induced analgesia and nocebo-induced hyperalgesia[12]. These circuits reflect the complex interplay between cognitive, emotional, and sensory systems in the modulation of pain, underscoring the necessity of cross-species studies to fully elucidate these processes.

Despite these advances, there has yet to be a direct comparison of the neurobiological mechanisms underlying placebo analgesia or nocebo hyperalgesia between humans and animals. Our study addresses this critical gap by directly comparing the regional activation, functional connectivity and neural circuits activated during placebo and nocebo responses in both humans and rats. By doing so, we aim to establish a cross-species framework that will enhance our understanding of conserved and species-specific mechanisms of pain modulation. This comparison is particularly important as it will validate rodent models as translational tools for pain research, facilitating the development of new interventions that target placebo and nocebo pathways.

## Results

### Response conditioning elicits placebo analgesia and nocebo hyperalgesia in both humans and rats

To elicit placebo analgesia and nocebo hyperalgesia in humans, we used a common response conditioning protocol. Low, moderate and high thermal stimuli were repeatedly paired with topical creams labelled Lidocaine (placebo), Vaseline (control) and Capsaicin (nocebo), respectively, on two consecutive days. Following this, placebo and nocebo responses were tested during fMRI scans, in which all three creams were reapplied, but the moderate intensity stimulus was delivered to each cream site (Fig. 1a–d). For humans, the 'control' therefore refers to the fMRI scan in which this moderate pain intensity was delivered to the Vaseline cream site. Permutation testing of pain ratings between the control cream and the Lidocaine or Capsaicin creams revealed placebo responder subpopulations (22/46; Fig. 1e–g) and nocebo responder subpopulations (14/25; Figure h–j), respectively. As previously reported[13,14], this conditioning protocol was effective at altered participants' expectations, as evidenced by significant differences in expected pain ratings for both the placebo and nocebo manipulations. There were no significant differences in expectations between placebo responders and non-responders, or between nocebo responders and non-responders, indicating that expectation changes occurred across all participants regardless of their behavioral response (Supplementary Table 1).

A similar protocol was used to elicit placebo analgesia and nocebo hyperalgesia in rats. However, while for the human study we were able to employ a within-subject design, the study in rats was necessarily a between-subjects design. As such, separate groups of rats underwent a response conditioning procedure in which either a low (placebo), moderate (control) or high (nocebo) intensity cold thermal stimulus was paired with contextual cues (Fig. 1k–m). To assess the impact of cue saliency on the strength of conditioned responses, placebo and nocebo groups were tested either in a 'minimal context' or an 'enhanced context' setting (Fig. 1n, o). Subsequently, all groups were tested in their same context at the moderate stimulus. For rats, the 'control group' therefore refers to the rats that were tested at the moderate intensity stimulus during both the conditioning procedure and on Test Day. Comparisons of pain behaviours between the control group and the low and high intensity conditioned groups revealed that this procedure produced strong placebo analgesia (Fig. 1p, q) and nocebo hyperalgesia (Fig. 1r, s) in the majority of rats tested.

Firstly, to assess changes in overall regional brain activity during placebo and nocebo in humans, beta values representing the degree of correlation between the blood oxygen level dependent (BOLD) fMRI signal intensity and the thermal stimulation pattern was calculated for 70 regions of interest (ROIs) for each fMRI scan (Fig. 2). Beta values of each ROI for the placebo/nocebo scans were then compared to the control scan. To assess changes in overall regional brain activity during placebo and nocebo in rats, c-Fos expression density in 70 ROIs across the brain was quantified for each placebo and nocebo group and compared to the control group. These ROIs were chosen based on their known or putative involvement in the processing, transmission and modulation of pain and/or contextual cues. Given

the different methodologies used to measure brain activity for humans and rats, to make direct comparisons between species we chose to use estimation statistics to identify the top 20 ROIs that showed the largest effect sizes between the placebo/nocebo conditions and controls. See Supplementary Data 2 for detailed results, and Supplementary Table 2 for complete list of ROIs and abbreviation.

### Human and rats show similar placebo-associated changes in activity across multiple brain regions

In humans, the average effect size (Cohen's d) between placebo and control for the top 20 ROIs was 0.36 (range 0.25–0.64). In rats, for the minimal context group, the average effect size in c-Fos expression density between placebo and control of the top 20 ROIs was 0.80 (range 0.53–2.14). For the enhanced context group, the average effect size of the top 20 ROIs was 0.64 (range 0.46–1.19). Similar patterns of activity were observed between the two rat groups. Placebo analgesia was associated with increases in c-Fos expression in 19/20 ROIs in the minimal context and 18/20 in the enhanced context, with 8 overlapping ROIs that showed similar effect sizes (the gracile nuclei, ACC, NAc, and basolateral amygdala [BLA]). However, the differences between these groups, most notably in the PAG, posterior insula cortex and paraventricular thalamus (PVT), likely reflects the effect of the enhanced context on regional brain activity. Remarkably, 8 regions were identified as having placebo-related changes in activity in both humans and rats. These were the contralateral primary somatosensory cortex (S1), ipsilateral cuneate/gracile, contralateral medial parabrachial nucleus (PB), ipsilateral basolateral amygdala, the septal nuclei and the rostral and caudal PVT (Fig. 3).

### Human and rats show similar nocebo-associated changes in activity across brain regions

In humans, the average effects size between the nocebo and control scans for the top 20 VOIs was 0.53 (range 0.39–0.95). In rats, for the minimal context group, the average effect size of the top 20 ROIs was 0.66 (range 0.49–1.12), while for the enhanced context group, the average effect size was 0.76 (range 0.6–1.15). Once again, similar overall patterns of activity were observed between these two rat groups, with 7 overlapping regions, each of which showed similar effect sizes. However, in stark contrast to the placebo groups, nearly all the top 20 ROIs in both nocebo groups had decreases in c-Fos expression compared to the control (16/20 in the minimal context and 19/20 in the enhanced context). Remarkably, 11 regions were identified as having nocebo-related changes in activity in both humans and rats. These were the ipsilateral primary motor cortex, caudal and rostral PVT, paraventricular hypothalamus, locus coeruleus, rostral ventromedial medulla, contralateral medial and lateral parabrachial nuclei, and the ipsilateral cuneate nucleus (Fig. 4).

### Functional connectivity and functional circuit networks

We next determined which neural circuits were active during placebo analgesia and nocebo hyperalgesia using functional connectivity. Functional connectivity can be defined as the temporal coincidence of spatially distant neurophysiological events[15]. That is, activity in one brain region/nucleus is correlated with and predictive of activity in another. For the rats, it is therefore possible to create functional connectivity maps for each group by correlating c-Fos expression between each of the ROIs. High or low correlations (Pearson's r-values) between two regions indicates high or low functional connectivity, respectively. Indeed, c-Fos functional connectivity is now a widely used and accepted approach[16–19].

In order to better align our human data with the rat analysis, we used the beta values from the fMRI analyses to assess functional connectivity, rather than utilizing traditional fMRI time-series data. As such, the beta values, representing the overall level of neural activation in each region during noxious stimuli, were correlated between ROIs to establish functional connectivity. This method provides a comparable measure to the c-Fos correlations used in rats, allowing us to assess functional relationships across species using a similar framework. To further facilitate cross-species comparisons, we identified the top 100 strongest functional connections from the

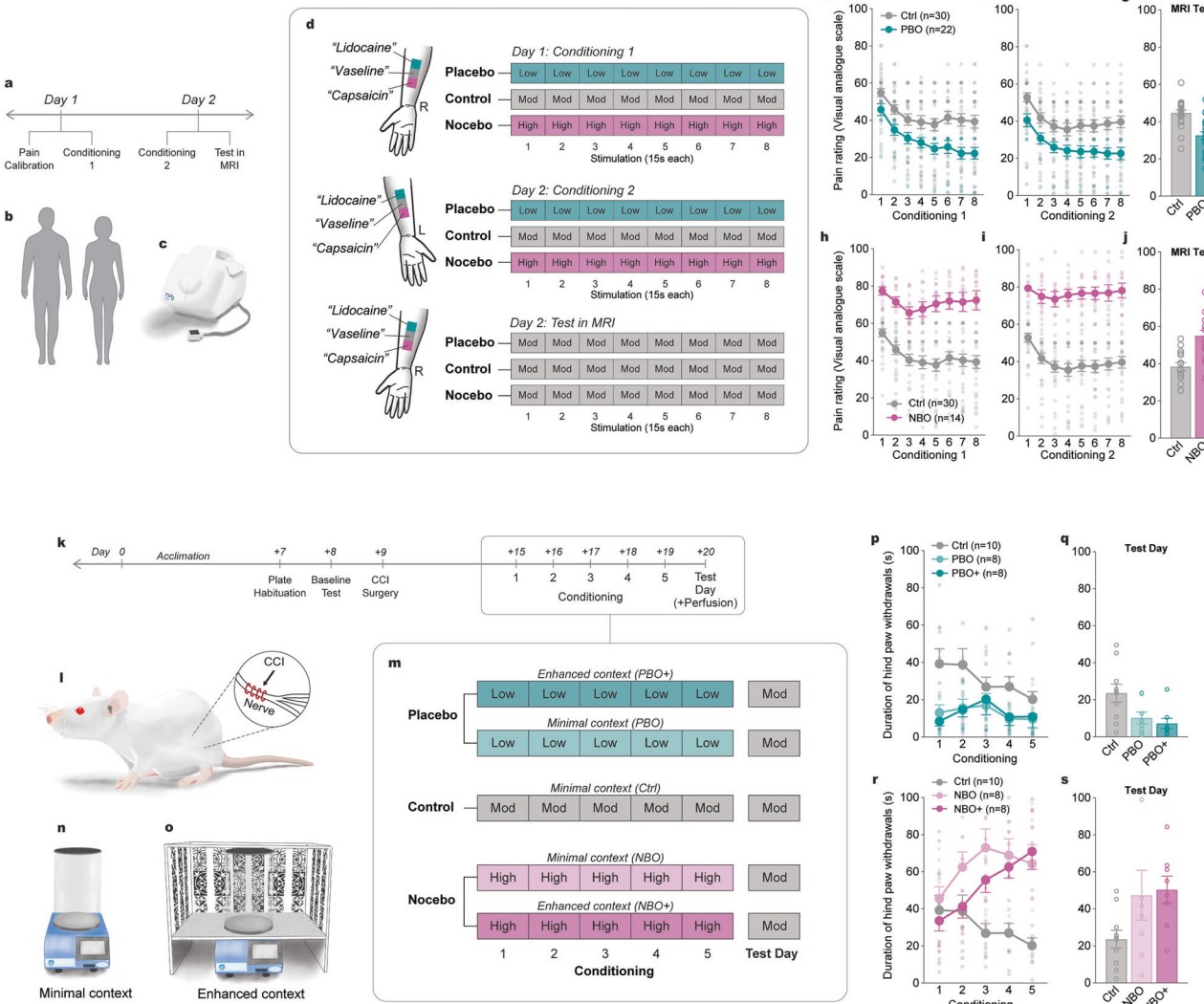

**Fig. 1 | Experimental design to elicit response-conditioned placebo analgesia and nocebo hyperalgesia in humans and rats. a, b** Experimental timeline for the human study. Healthy male and female participants underwent a two-day response-conditioning procedure to thermal stimuli using a MRI-compatible thermode (**c**). On day 1, participants first underwent a pain calibration assay to determine the temperatures that elicited a low, moderate and high pain rating for each individual. Following this, participants underwent round 1 of conditioning. On day 2, participants received round 2 of conditioning on the contralateral arm, followed by placebo and nocebo testing in an MRI scanner. **d** To condition placebo and nocebo responses, three topical creams were applied to the volar forearm and heat thermal stimuli (8 x 15s) applied to the cream sites. Participants were told that one of the creams ("Lidocaine") was a powerful analgesic, one of the creams ("capsaicin") was a powerful hyperalgesic, and one of the creams ("vaseline") was a control cream, which did not modify pain. However, none of these creams contained any active ingredients. To condition placebo and nocebo responses respectively, each time the thermode was placed on the "Lidocaine" cream the temperature of the thermode was surreptitiously reduced, while the temperature was surreptitiously increased when placed on the "capsaicin" cream. During the Test in the MRI, all three cream sites received the moderate pain temperature. **e, f** During conditioning, participants rated the pain on the "Lidocaine" cream lower than the control cream, and **h, i** rated the pain on the "capsaicin" cream as higher than the control cream. **g** During the Test in

the MRI, participants continued to rate the pain on "Lidocaine" cream site as lower than the control cream site, indicating placebo analgesia, while **j** participants continued to rate the pain on the "capsaicin" cream site as higher than the control cream site, indicating nocebo hyperalgesia. **k, l** Experimental timeline for the rat study. Male, Sprague-Dawley rats all received a unilateral chronic constriction injury of the sciatic nerve, a model of persistent neuropathic pain known to elicit strong cold allodynia. Six days later, they underwent a 5-day response-conditioning procedure to cold thermal stimuli using a hot/cold plate analgesiometer. **m** During conditioning, separate groups of rats were tested twice daily using either a low pain stimulus (30 °C - placebo), a moderate pain stimulus (20 °C - control) or a high pain stimulus (4 °C - nocebo). Three groups of rats were tested in (**n**) a minimal context setting while two groups of rats were testing in (**o**) an enhanced context setting, which also included strong visual, olfactory and auditory cues. The following day, on Test Day, all 5 groups of rats were tested at the moderate pain stimulus. **p** During conditioning, rats tested at the low pain stimulus demonstrated lower injured hind paw withdrawal behaviour than control rats, while (**r**) rats tested at the high pain stimulus demonstrated greater hind paw withdrawal behaviours. **q** On Test Day, rats conditioned to the low pain stimulus continued to demonstrate low reduced hind paw withdrawals, indicating placebo analgesia, while (**s**) rats tested at the high pain stimulus demonstrated increased hind paw withdrawals, indicating nocebo hyperalgesia. Error bars indicate SEM.

control group data in both rats and humans. These connections were selected based on their strength of correlation (Pearson's r values) and plotted to visualize and compare connectivity patterns between the two species (Supplementary Fig. 1). Comparison of functional connectivity maps in the control groups revealed species-specific patterns. The top 100 human

connections were dominated by left-right frontal cortical links, while rats exhibited stronger cortical-subcortical connectivity. Notably, 16 connections were shared between species, significantly higher than the 4.66 expected by chance, largely involving frontal cortical regions, the ACC, and middle cingulate cortex (Table 1). Graph metrics were comparable: the rat map

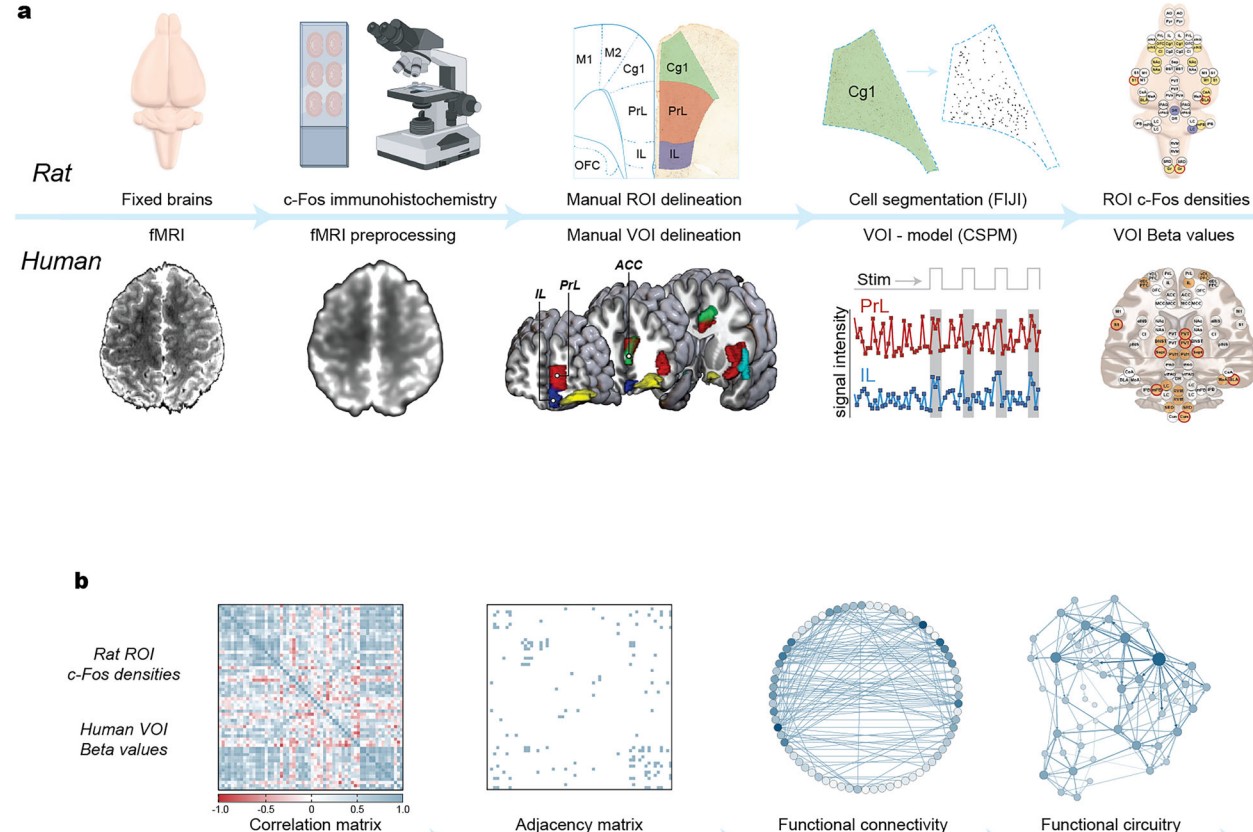

**Fig. 2 | Methods to quantify brain activity in rats and humans to identify the functional circuitry underlying placebo analgesia and nocebo hyperalgesia.**
**a** Pipelines for quantifying regional neural activity in rats and humans. For rats, c-Fos immunohistochemistry was used to quantify neural activity across 70 regions of interest (ROIs). ROIs were manually delineated based on the Paxinos and Watson (2005) atlas, and c-Fos densities were calculated using ImageJ by segmenting and automatically counting stained cells within each ROI. For humans, fMRI data were preprocessed using SPM12. ROIs were manually delineated, and beta values were calculated from the BOLD signal intensity changes, relative to baseline period at the start of the scan. **b** Pipeline for assessing functional connectivity and functional

circuitry during control, placebo, and nocebo tests. For both the c-Fos densities in rats and the beta values in humans, correlation matrices were generated for each group, which measures the strength and direction of the correlations between each ROI pair. Adjacency matrices were constructed for each group, based on the top 100 strongest correlations in the control group. Functional connectivity maps were created using Gephi software. The resulting functional connections were then pruned based on neuroanatomical tract tracing data catalogued by the Rat Connectome Project of the University of Rostock, and force-directed graphs of the resulting functional circuitry were produced using Gephi. BioRender images used in the creation of this Figure.

showed an average path length of 3.145, the human map had an average path length of 3.479, reflecting similar network efficiency despite species-specific topologies (see Supplementary Table 3 for full graph metrics).

## Enhancing the context during conditioning increases brain-wide functional connectivity

To assess changes in functional connectivity during placebo and nocebo responses, we applied the same $r$-value cut-offs as used in the control groups (rats: $r = 0.879$, humans: $r = 0.699$) (See Supplementary Fig. 4), with connections shared with controls excluded from this analysis to isolate only unique placebo-related and nocebo-related connections. For the rats, this analysis revealed a divergent pattern: minimal context rats exhibited a decrease or no change to their functional connectivity, with 69 connections for placebo and 105 for nocebo, whereas the enhanced context placebo and nocebo groups had 291 and 268 connections, respectively (Supplementary Figs. 2, 3). This would indicate that the enhanced context greatly increased overall functional connectivity of the brain, particularly between frontal cortical regions and the amygdala and olfactory areas.

## Conserved functional connectivity patterns in human and rat placebo responses

Human placebo responses also showed an increase in overall functional connectivity, expanding to 200 connections compared to controls. Notably,

47 of these connections were shared between humans and the two rat placebo groups (Table 2; Supplementary Fig. 2). This shared connectivity was concentrated in the amygdala, ACC, and NAc. These overlaps suggest a conserved set of circuits or networks across species that either accompany or drive the placebo analgesic response.

## Limited cross-species overlap in nocebo-related functional connectivity

Human nocebo responses showed a modest increase in overall functional connectivity, with 130 connections compared to controls. In contrast to the placebo condition, only 4 connections were shared between humans and minimal context rats, and 19 with enhanced context rats. The increased connectivity in enhanced context rats (268 connections), did not translate to significant overlap with humans. Importantly, the nocebo networks exhibited a distinct topology from placebo networks, particularly with reduced amygdala involvement in human nocebo responses. This suggests that nocebo hyperalgesia is driven by different neural circuits across species, highlighting the specificity of nocebo-related mechanisms.

## Neural circuits underlying pain, placebo analgesia, and nocebo hyperalgesia

A major caveat of functional connectivity analysis is that it only reflects statistical correlations, not direct anatomical or causal relationships between

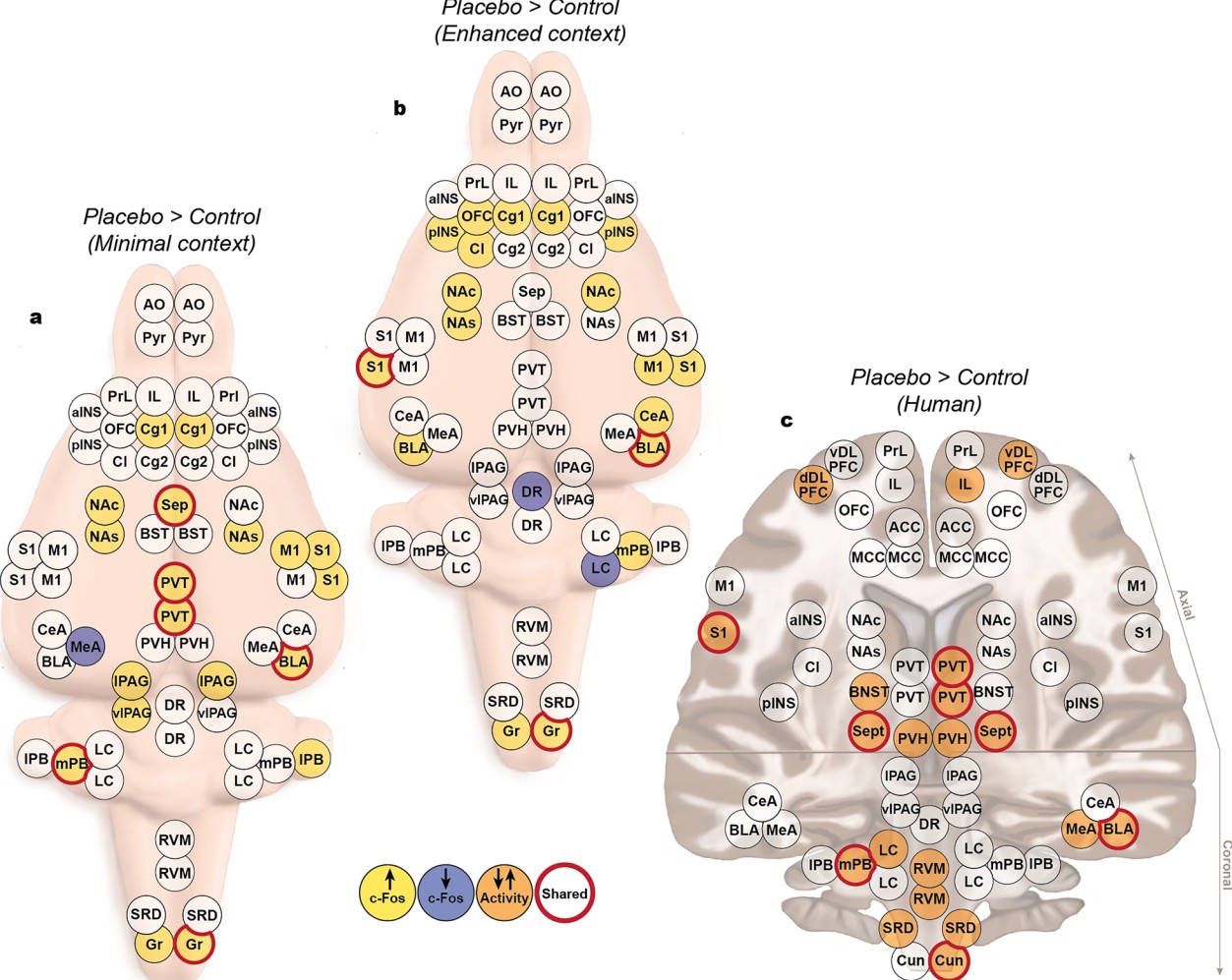

**Fig. 3 | Changes in brain activity during placebo analgesia.** The top 20 ROIs that showed increased or decreased c-Fos expression compared to controls for rats tested in the (**a**) minimal context and (**b**) enhanced context. **c** The top 20 ROIs that showed increased or decreased beta values compared to control scan in humans. Red circles indicate the ROIs that showed activity changes in both rats and humans. AO anterior olfactory tubercle, Pyr pyriform cortex, aINS anterior insula, pINS posterior insula cortex, PrL prelimbic cortex, IL infralimbic cortex, OFC orbitofrontal cortex, Cg1 cingulate cortex, Cg2 cingulate cortex, Cl claustrum, Sep septal nuclei, BST bed nucleus of the stria terminalis, NAc nucleus accumbens core, NAs nucleus accumbens shell, M1 primary motor cortex, S1 primary somatosensory cortex, PVT paraventricular thalamus, PVH paraventricular hypothalamus, CeA central nucleus of the amygdala, MeA medial nucleus of the amygdala, BLA basolateral nucleus of the amygdala, lPAG lateral periaqueductal gray, vlPAG ventrolateral periaqueductal gray, DR dorsal raphe, LC locus coeruleus, mPB medial parabrachial nucleus, lPB lateral parabrachial nucleus, RVM rostral ventromedial medulla, SRD subnucleus reticularis dorsalis, Gr gracile nucleus, ACC anterior cingulate cortex, MCC middle cingulate cortex, Cun cuneate nucleus.

regions. To refine our analysis, we pruned the functional connections using anatomical data from the Rat Connectome Project of the University of Rostock[20], which compiles tract-tracing studies in rodents. From this database, we extracted projection direction and strength for every functional connection in the isolated networks shown in Supplementary Figs. 2–4. If there was weak or no evidence for an anatomical connection, it was removed from the network.

We first applied this pruning and assigned strength to the control groups, generating force-directed graphs for humans and rats (Fig. 5). Shared circuits are highlighted in red. Both networks showed similar overall patterns, with frontal cortical regions interconnected in both species, while subcortical areas like the PAG and other brainstem regions had primarily local connections. The overlapping circuits between species were related predominantly to prefrontal and PAG areas. Indeed, despite the differences in these control networks, the presence of conserved pathways suggests fundamental similarities in pain processing between humans and rats.

Figure 6 shows the pruned placebo networks for humans and rats, and all shared connections are detailed in Table 2, which includes their direction

and the strength of these pathways. Several key hub regions, defined as those with the highest weighted degree of connectivity, emerged across species and conditions. In humans, the PAG, posterior insular cortex, ACC, and BLA served as the primary hubs. The rat minimal context network, while more modular, shared similar hub regions, including the BLA, anterior insula, ACC, vlPAG, and orbitofrontal cortex. Interestingly, while the PAG in the minimal context rats formed an isolated, island-like network disconnected from the broader network, the human PAG was more integrated, with connections extending to other major hubs. In the enhanced context rats, hub activity shifted to regions such as the anterior and posterior insula, pyriform cortex, orbitofrontal cortex, and prelimbic cortex. The appearance of the pyriform cortex, anterior olfactory nucleus, and medial amygdala in this context likely reflects processing of the strong olfactory cues used during conditioning. Despite these differences, several core connections are shared across species, particularly in prefrontal and pain-modulatory regions, suggesting conserved pathways in placebo analgesia.

Figure 7 shows the pruned nocebo networks for humans and rats, and all shared connections for nocebo groups are detailed in Table 3, which

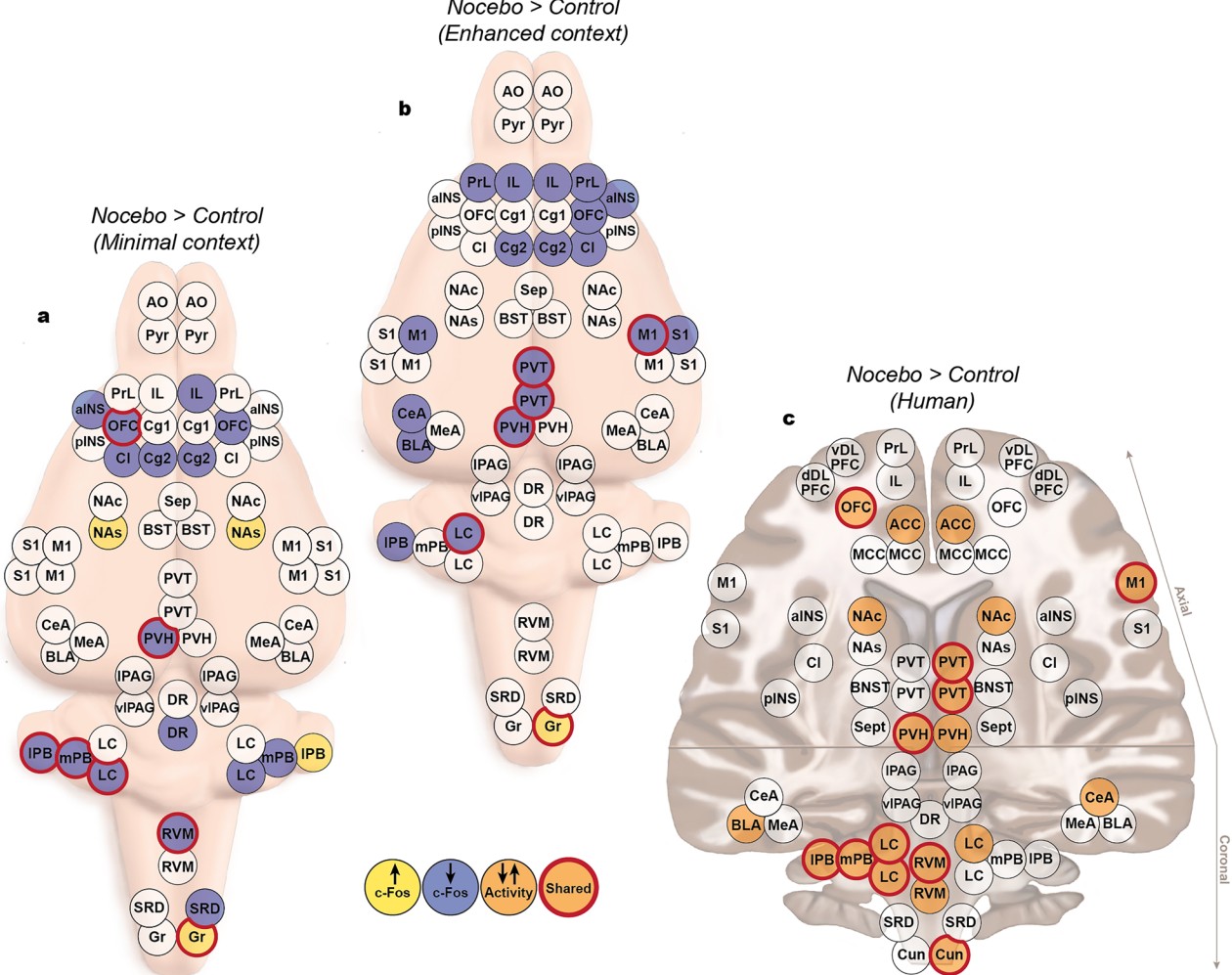

**Fig. 4 | Changes in brain activity during nocebo hyperalgesia.** The top 20 ROIs that showed increased or decreased c-Fos expression compared to controls for rats tested in the (**a**) minimal context and (**b**) enhanced context. **c** The top 20 ROIs that showed increased or decreased beta values compared to control scan in humans. Red circles indicate the ROIs that showed activity changes in both rats and humans. AO anterior olfactory tubercle, Pyr pyriform cortex, aINS anterior insula, pINS posterior insula cortex, PrL prelimbic cortex, IL infralimbic cortex, OFC orbitofrontal cortex, Cg1 cingulate cortex, Cg2 cingulate cortex, Cl claustrum, Sep septal nuclei, BST bed nucleus of the stria terminalis, NAc nucleus accumbens core, NAs nucleus accumbens shell, M1 primary motor cortex, S1 primary somatosensory cortex, PVT paraventricular thalamus, PVH paraventricular hypothalamus, CeA central nucleus of the amygdala, MeA, medial nucleus of the amygdala, BLA basolateral nucleus of the amygdala, lPAG lateral periaqueductal gray, vlPAG ventrolateral periaqueductal gray, DR dorsal raphe, LC locus coeruleus, mPB medial parabrachial nucleus, lPB lateral parabrachial nucleus, RVM rostral ventromedial medulla, SRD subnucleus reticularis dorsalis, Gr gracile nucleus, ACC anterior cingulate cortex, MCC middle cingulate cortex, Cun cuneate nucleus.

includes their direction and the strength of these pathways. Key hub regions were identified in both humans and rats, though with fewer shared connections compared to placebo. In humans, the primary hubs included the prelimbic cortex, motor cortex, anterior and middle cingulate cortex, and the PAG. Unlike the placebo network, the PAG in humans formed an isolated network, showing no integration with the broader nocebo network. In the rat minimal context, the main hubs were the prelimbic cortex, posterior insula, and ACC, but notably, there were no shared connections between this network and the human nocebo network. For the enhanced context, the primary hubs shifted to the anterior insula, ACC, vlPAG, and BLA. Although there were more shared connections between the enhanced context rats and humans, the overall overlap was still far less than that observed in the placebo networks. These differences, particularly the isolated PAG in humans and the more integrated PAG in the enhanced context rats, highlight distinct network organizations between placebo and nocebo conditions. The shared connections across species were limited, suggesting a more species-specific recruitment of circuits during nocebo hyperalgesia compared to placebo analgesia.

## Discussion

This study presents a novel approach to cross-species analysis, utilizing c-Fos immunohistochemistry in rats and fMRI in humans to create comparable, species-specific maps of brain activity and neural circuitry. We propose an innovative framework for investigating conserved and distinct brain circuits underlying complex behaviours and physiological responses. Furthermore, this approach enables preclinical research to prioritize the neural pathways that are most likely conserved between species, accelerating translational efforts and refining focus on the circuits with the greatest relevance to humans. With respect to placebo analgesia and nocebo hyperalgesia, our findings revealed that while there are notable species-specific differences, certain key regions—such as the ACC, PAG, and amygdala—show conserved patterns of activation and connectivity in response to placebo and nocebo conditioning. However, given the novelty of our approach, when interpreting these findings, two critical considerations arise: 1) how well our results align with established literature on placebo and nocebo responses, and 2) the comparability of the methodologies used for each species.

**Table 1 | Shared ROI-ROI functional and anatomical connections between humans and rats in the control groups**

| Functional Connection | Pearson's r | | Anatomical connection | Strength(s) |
| --- | --- | --- | --- | --- |
| | Human | Rat (minimal) | | |
| PrL(L)-PrL(R) | 0.890 | 0.972 | PrL(L)←—→PrL(R) | 3, 3 |
| PrL(L)-ACC1(R) | 0.702 | 0.966 | PrL(L)→ACC1(R) | 1 |
| PrL(L)-ACC1(L) | 0.705 | 0.952 | PrL(L)→ACC1(L) | 1 |
| IL(L)-IL(R) | 0.877 | 0.933 | IL(L)←—→IL(R) | 3, 3 |
| S1(L)-M1(L) | 0.836 | 0.879 | S1(L)→M1(L) | 3 |
| ACC1(L)-ACC1(R) | 0.931 | 0.975 | - | |
| ACC2(L)-ACC1(R) | 0.876 | 0.939 | - | |
| ACC2(L)-ACC2(R) | 0.931 | 0.906 | ACC2(L)←—→ACC2(R) | 2, 2 |
| MCC1(L)-MCC1(R) | 0.966 | 0.932 | - | |
| MCC2(L)-MCC2(R) | 0.973 | 0.959 | - | |
| rVLPAG(L)-rLPAG(R) | 0.704 | 0.934 | rVLPAG(L)←—→rLPAG(R) | 3, 0.5 |
| cVLPAG(L)-cDR | 0.887 | 0.903 | cVLPAG(L)←—→cDR | 3, 3 |
| rLPAG(L)-rVLPAG(R) | 0.863 | 0.883 | rLPAG(L)←—→rVLPAG(R) | 0.5, 3 |
| PVH(L)-PVH(R) | 0.871 | 0.888 | PVH(L)←—→PVH(R) | 1, 1 |
| PrL(R)-ACC1(R) | 0.710 | 0.913 | PrL(R)→ACC1(R) | 3 |
| S1(R)-M1(R) | 0.903 | 0.905 | S1(R)→M1(R) | 3 |

Arrows indicate the direction(s) of the anatomical connections.

## How do our findings compare to what is known about the neural activity during placebo analgesia and nocebo hyperalgesia?

fMRI has now been extensively employed to investigate the neural basis of placebo analgesia, and, to a lesser extent, nocebo hyperalgesia. However, while many of these studies often identify different collections of brain regions responsive during placebo and nocebo manipulations, several key regions consistently emerge central to these responses. Notably, the meta-analysis by Zunhammer et al.[21] synthesized data from 603 participants across 20 fMRI studies, revealing common areas of activation associated with placebo analgesia. These included the mid-cingulate and insula cortices, frontoparietal regions and the thalamus. In our study, despite our different analysis pipeline —using beta values within 70 pre-defined regions of interest rather than a whole-brain voxel-based analysis—we observed activation patterns that generally align with previous research.

The top 20 brain regions in humans that showed the largest changes during placebo analgesia in our study (see Fig. 3) fell into three categories. Firstly, regions that have previously been shown to be involved in placebo analgesia. These were the majority of the regions identified by our analysis: the dorsolateral prefrontal cortex (dlPFC[22,23]), the medial prefrontal cortex (mPFC[24]), the rostral ventromedial medulla (RVM), para-ventricular hypothalamus (PVH[25]), the amygdala[26], S1, and the thalamus[21]. Secondly, brainstem regions that are known to be involved in pain modulation but currently have scarce direct evidence for their involvement in placebo analgesia (see ref. 27 for review). These were the medial PB (mPB), the locus coeruleus (LC), the subnucleus reticularis dorsalis (SRD) and the cuneate nucleus ipsilateral to the stimulus. Thirdly, regions that have not been previously associated with placebo analgesia. These were the septal nuclei and the bed nucleus of the stria terminalis (BNST).

Similarly, the top 20 brain regions in humans that showed the largest changes during nocebo hyperalgesia (see Fig. 4) fell into the same three categories. While considerably fewer studies have investigated the neural correlates of nocebo hyperalgesia, half of the regions identified by our analysis have previously been reported. These were the orbitofrontal cortex (OFC[9,28]; Schiene et al. 2018), ACC[29]), NAc, and PVH[9]. Several pain modulatory brainstem regions were also identified. These were the LC, RVM, medial and lateral PB, and the cuneate nucleus. Finally, the regions identified by our analysis that have yet to be associated with

nocebo hyperalgesia were specifically the primary motor cortex (M1), PVT, BLA and the CeA.

With respect to preclinical research, only a handful of studies have explored the neurobiological correlates of placebo analgesia, yet once again, several brain regions have emerged as central. For instance, the rostral rACC[10,30]), ventral tegmental area (VTA[30]), ventrolateral PAG (vlPAG), medial prefrontal cortex, NAc[7,11]) have all been shown to have altered activity during the expression of placebo analgesia. Additionally, a recent study by Chen et al. (2024)[6] identified an ACC-pontine circuit projecting into the cerebellum as both necessary and sufficient for producing placebo analgesia; while Chen et al. (2024)[5] found that direct activation of central amygdala (CeA) neurons can condition placebo responses, but these neurons do not re-activate during the expression of placebo analgesia. Our study identified many of these same regions as having large changes in neural activity, including the vlPAG, ACC, CeA, and NAc, highlighting a remarkable consistency with this limited existing literature. It is also worth noting that many regions included in our study—such as specific brainstem and subcortical nuclei—have not been previously investigated in animal models of placebo analgesia. While there have been three recent studies that have developed animal models of nocebo hyperalgesia[4,31,32], to our knowledge this is the first study to investigate the neural basis of these effects. However, interestingly Zhang et al. (2024)[33], also recently found that the ACC, BLA, and PVT had increased c-Fos expression in their rat model of nocebo nausea.

Finally, our findings reveal large and important differences in both the regional activity and the functional connectivity between placebo and nocebo conditions, with distinct networks emerging for each. The circuits recruited in placebo and nocebo show minimal overlap, suggesting they rely on fundamentally different neural mechanisms. In other words, placebo analgesia and nocebo hyperalgesia do not merely represent opposite shifts in activity within a shared pain circuit, but rather engage distinct and functionally separate neural pathways as previously suggested[9]. Additionally, the minimal overlap in functional connectivity between rats and humans in the nocebo condition suggests greater species-specific variability in how the nocebo response conditioning influences pain processing. This underscores the need for future research to investigate placebo and nocebo as fundamentally distinct phenomena to better understand their unique mechanisms and develop more precise therapeutic strategies.

**Table 2 | Shared ROI-ROI functional and anatomical connections between humans and rats in the placebo groups**

| Functional Connection | Pearson's r | | | Anatomical connection | Strength(s) |
|---|---|---|---|---|---|
| | Human | Rat (min) | Rat (enh) | | |
| PrL(R)-ACC-1(R) | 0.705 | 0.765 | 0.949 | PrL(R)←→ACC-1(R) | 1, 3 |
| pINS(R)-ACC-2(R) | 0.798 | 0.268 | 0.909 | pINS(R)←→ACC-2(R) | 1, 1 |
| pINS(R)-MCC-1(R) | 0.787 | 0.749 | 0.884 | pINS(R)→MCC-1(R) | 2 |
| pINS(R)-MCC-2(R) | 0.714 | 0.606 | 0.911 | pINS(R)←MCC-2(R) | 3 |
| pINS(R)-pINS(L) | 0.789 | 0.714 | 0.958 | pINS(R)←→pINS(L) | 3, 3 |
| pINS(R)-aINS(L) | 0.756 | 0.869 | 0.908 | pINS(R)←→aINS(L) | 3, 3 |
| pINS(R)-ACC-2(L) | 0.718 | 0.339 | 0.898 | pINS(R)←→ACC-2(L) | 3, 1 |
| pINS(R)-MCC-1(L) | 0.769 | 0.305 | 0.935 | - | |
| pINS(R)-MCC-2(L) | 0.727 | 0.561 | 0.923 | - | |
| pINS(R)-rLPAG(L) | 0.772 | 0.904 | −0.073 | pINS(R)→rLPAG(L) | 1 |
| ACC-1(R)-Claus(L) | 0.785 | 0.764 | 0.881 | ACC-1(R)←→Claus(L) | 2, 3 |
| ACC-1(R)-MCC-1(L) | 0.833 | 0.922 | 0.868 | - | |
| ACC-1(R)-MCC-1(R) | 0.807 | 0.709 | 0.951 | ACC-1(R)←→MCC-1(R) | 3, 1 |
| ACC-2(R)-ACC-1(L) | 0.872 | 0.902 | 0.808 | ACC-2(R)←ACC-1(L) | 2 |
| ACC-2(R)-MCC-1(L) | 0.765 | 0.834 | 0.966 | - | |
| ACC-2(R)-MCC-2(L) | 0.866 | 0.698 | 0.899 | - | |
| NAc(R)-NAc(L) | 0.858 | 0.514 | 0.974 | - | |
| NAc(R)-Nash(L) | 0.823 | 0.594 | 0.895 | - | |
| NAc(R)-CeA(L) | 0.728 | −0.116 | 0.946 | - | |
| BLA(R)-NAc(L) | 0.755 | 0.477 | 0.920 | - | |
| BLA(R)-Nash(L) | 0.784 | 0.456 | 0.935 | BLA(R)←→Nash(L) | 3, 3 |
| BLA(R)-BLA(L) | 0.904 | 0.952 | 0.877 | BLA(R)←→BLA(L) | 1, 1 |
| BLA(R)-CeA(L) | 0.811 | 0.619 | 0.957 | - | |
| MeA(R)-Sept(L) | 0.706 | 0.372 | 0.953 | MeA(R)←→Sept(L) | 1.5, 2 |
| MeA(R)-MeA(L) | 0.759 | 0.002 | 0.914 | - | |
| rVPLAG(R)-rVPLAG(L) | 0.896 | 0.918 | 0.864 | rVPLAG(R)←→rVPLAG(L) | 3, 3 |
| rVPLAG(R)-rLPAG(R) | 0.949 | 0.409 | 0.908 | rVPLAG(R)←→rLPAG(R) | 3, 0.5 |
| cVPLAG(R)-cDR | 0.843 | 0.521 | 0.889 | - | |
| rLPAG(R)-cLPAG(R) | 0.702 | 0.792 | 0.978 | rLPAG(R)←→cLPAG(R) | 2, 2 |
| cLPAG(R)-cVLPAG(L) | 0.750 | 0.898 | 0.943 | cLPAG(R)←→cVLPAG(L) | 0.5, 3 |
| cLPAG(R)-cLPAG(L) | 0.811 | 0.771 | 0.967 | cLPAG(R)←→cLPAG(L) | 2, 2 |
| cLPAG(L)-rLPAG(L) | 0.708 | 0.624 | 0.899 | cLPAG(L)←→rLPAG(L) | 2, 2 |
| rLC(R)-rLC(L) | 0.852 | 0.735 | 0.972 | rLC(R)←→rLC(L) | 3, 3 |
| cLPAG(L)-cVLPAG(L) | 0.901 | 0.832 | 0.987 | cLPAG(L)←→cVLPAG(L) | 3, 3 |
| CeA(L)-Nash(L) | 0.753 | 0.301 | 0.933 | CeA(L)→Nash(L) | 1.5 |
| CeA-(L)-NAc(L) | 0.721 | 0.112 | 0.974 | CeA-(L)←NAc(L) | 0.5 |
| Sept(L)-Nash(L) | 0.791 | 0.684 | 0.887 | Sept(L)→Nash(L) | 2 |
| Nash(L)-NAc(L) | 0.914 | 0.824 | 0.908 | Nash(L)←→NAc(L) | 2, 2 |
| Nash(L)-PrL(L) | 0.747 | 0.909 | 0.500 | Nash(L)←PrL(L) | 3 |
| MCC-1(L)-MCC-2(L) | 0.890 | 0.741 | 0.905 | MCC-1(L)←→MCC-2(L) | 4, 3 |
| ACC-1(L)-ACC-2(L) | 0.896 | 0.957 | 0.937 | ACC-1(L)←→ACC-2(L) | 0.5, 1 |
| ACC-2(L)-PrL(L) | 0.789 | 0.874 | 0.903 | ACC-2(L)←→PrL(L) | 1, 2 |
| PrL(L)-IL(L) | 0.759 | 0.640 | 0.976 | PrL(L)←→IL(L) | 3, 4 |
| OFC(L)-Claus(L) | 0.721 | 0.850 | 0.972 | OFC(L)←→Claus(L) | 3, 2 |
| lPB(L)-mPB(L) | 0.892 | 0.794 | 0.884 | - | |
| rPVT(R)-cPVT(R) | 0.821 | 0.320 | 0.883 | - | |
| rPVT(L)-cPVT(L) | 0.743 | 0.320 | 0.883 | - | |

Arrows indicate the direction(s) of the anatomical connections.

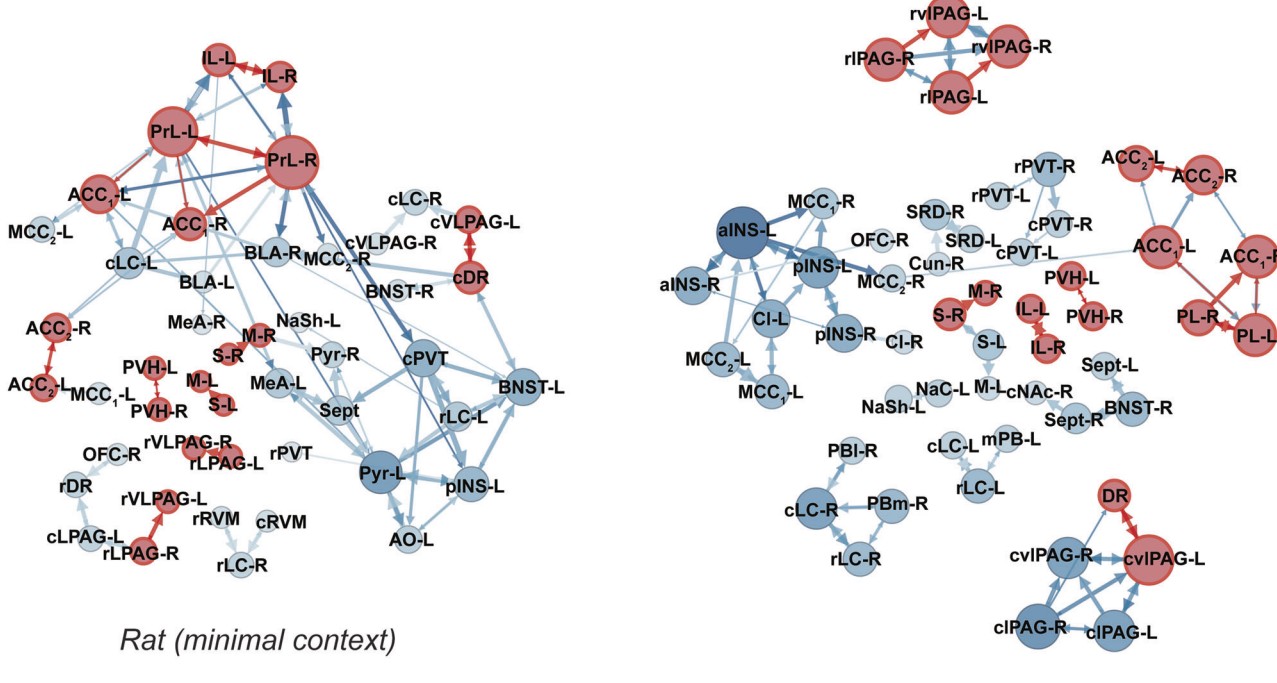

*Rat (minimal context)*

*Human*

**Fig. 5 | Functional neural circuitry of control groups (moderate pain control test or scan).** Functional connectivity maps between all ROIs were first generated using c-Fos expression in rats and beta-values in humans, with the top 100 strongest connections identified (Supplementary Fig. 1). The direction and strength of each of these connections was then applied based on known anatomical projections (Rat Connectome Project), or pruned if there was no evidence of a projection, and force directed graphs were created using Gephi. The size of the nodes reflect the degree of connectivity in the network. The size of the arrows indicate the strength of the projection. Red nodes and arrows indicate shared connections between rats and humans. AO anterior olfactory tubercle, Pyr pyriform cortex, aINS anterior insula, pINS posterior insula cortex, PrL prelimbic cortex, IL infralimbic cortex, OFC orbitofrontal cortex, ACC1 dorsal anterior cingulate cortex, ACC2 ventral anterior cingulate cortex, Cl claustrum, Sept septal nuclei, BNST bed nucleus of the stria terminalis, NAc nucleus accumbens core, NaSh nucleus accumbens shell, M1 primary motor cortex, S1 primary somatosensory cortex, PVT paraventricular thalamus, PVH paraventricular hypothalamus, CeA central nucleus of the amygdala, MeA medial nucleus of the amygdala, BLA basolateral nucleus of the amygdala, lPAG lateral periaqueductal gray, vlPAG ventrolateral periaqueductal gray, DR dorsal raphe, LC locus coeruleus, mPB medial parabrachial nucleus, lPB lateral parabrachial nucleus, RVM rostral ventromedial medulla, SRD subnucleus reticularis dorsalis, Gr gracile nucleus, ACC anterior cingulate cortex, MCC1 dorsal middle cingulate cortex, MCC2 ventral middle cingulate cortex, Cun cuneate nucleus.

## Methodological comparability: bridging human and animal models

Comparing neural activity across humans and rats in the context of placebo and nocebo effects introduces multiple layers of complexity. To meaningfully interpret our findings, we must consider the comparability of our approaches across three key levels: (i) the behavioural protocols used to induce placebo and nocebo responses in each species, (ii) the methodologies employed to assess neural activity (i.e., c-Fos immunohistochemistry in rats and fMRI in humans), and (iii) the analytical techniques applied to extract functional connectivity and circuit-level insights from these data.

At the behavioural model level, both human and rodent paradigms employ response conditioning, in which the intensity of a thermal stimulus was repeatedly paired with contextual cues to produce learned associations. While there are obvious differences between the procedures (for instance, in humans these associations are reinforced with verbal instructions), ultimately both paradigms are designed to produce expectancies and predictions about an upcoming stimulus. It therefore seems likely that each would engage the same neural mechanisms, and indeed the limited research in rodents suggests this is the case[6,7]. Further, the inclusion of the enhanced context group was designed to capture the multisensory cues inherent in human conditioning protocols. By incorporating a combination of olfactory, auditory, and visual stimuli, this group aimed to amplify the salience of the contextual pairings, thereby mimicking the cognitive and sensory complexity of human expectancy formation.

One notable difference between the human and rodent models is that the rats were given a CCI while the human participants were healthy and uninjured. Importantly, for our purposes, both human and rodent paradigms used in our study relied on acute stimulus-evoked responses rather than spontaneous pain, making them comparable in assessing placebo- and nocebo- induced pain modulation. However, it remains possible that the CCI model alters neural processing in response to pain, placebo, and nocebo, which could introduce differences in activation patterns compared to healthy individuals. While there has yet to be an animal study that investigates whether this is the case, Wang et al. (2022)[34] demonstrated that participants with chronic neuropathic pain (TMD) exhibit comparable levels of placebo analgesia to healthy individuals, indicating that the neural circuitry underlying acute placebo responses remains intact in people despite the presence of chronic pain. This suggests that while chronic pain conditions may influence certain aspects of pain processing, they do not necessarily impair the fundamental mechanisms driving placebo effects. Given these considerations, it would be worthwhile to replicate our findings using uninjured rodent models, particularly since recent studies have developed more directly comparable models of placebo analgesia and nocebo hyperalgesia in healthy mice[35,36]. Expanding research to include both injured and uninjured models will also help clarify the extent to which chronic pain state influences placebo and nocebo mechanisms and enhance the generalizability of these findings.

At the methodological level, *prima facie*, c-Fos expression and fMRI appear to be fundamentally different approaches, with c-Fos expression being a cellular-level marker and fMRI being a regional, hemodynamic-based measurement. Additionally, while fMRI offers essentially a real-time measure of regional activity, the delayed nature of c-Fos production

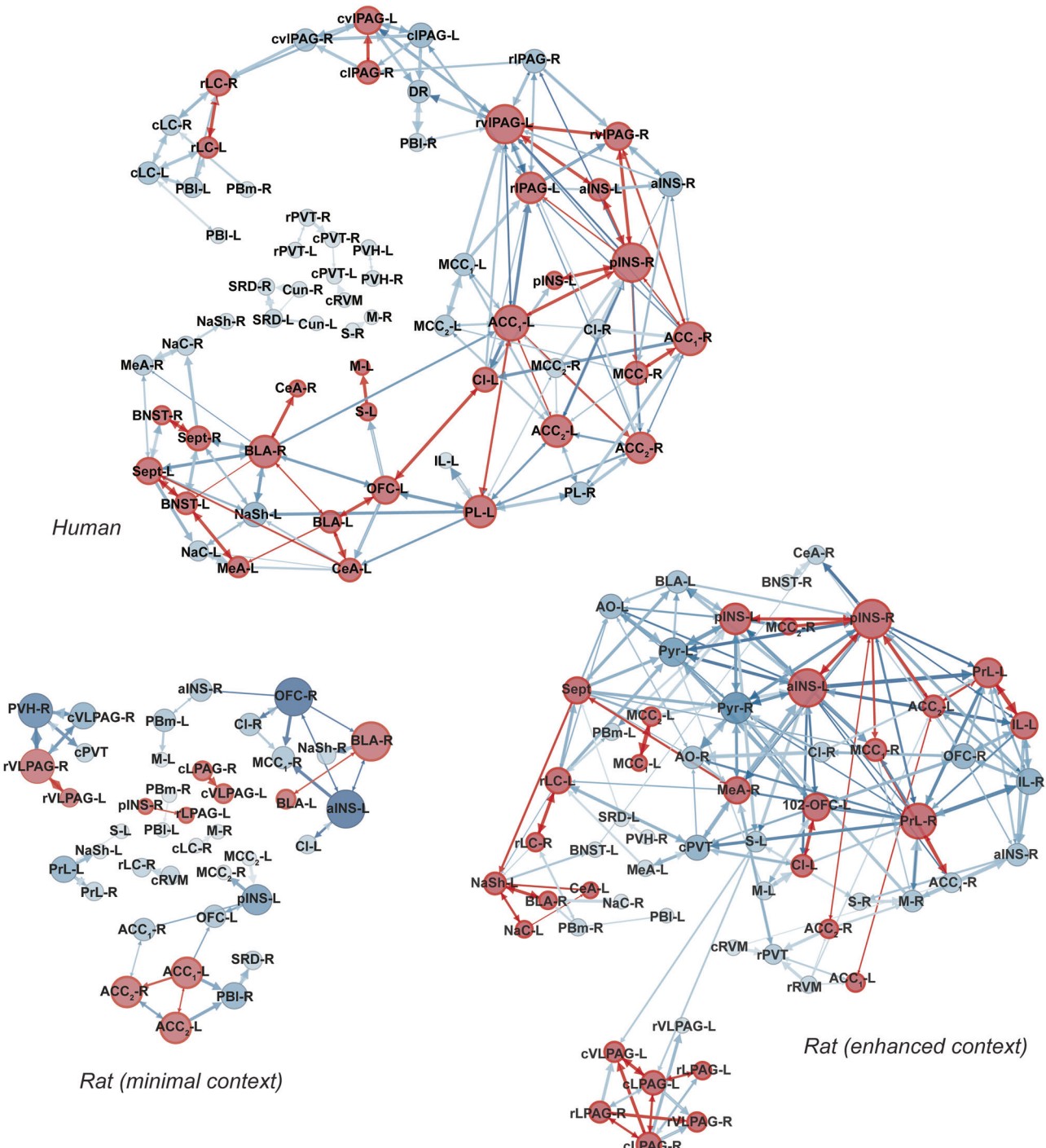

**Fig. 6 | Functional neural circuitry during placebo analgesia.** Functional connectivity maps between all ROIs were first generated using c-Fos expression in rats and beta-values in humans, with the top 100 strongest connections identified (Supplementary Fig. 1). The direction and strength of each of these connections was then applied based on known anatomical projections (Rat Connectome Project), or pruned if there was no evidence of a projection, and force directed graphs were created using Gephi. The size of the nodes reflect the degree of connectivity in the network. The size of the arrows indicate the strength of the projection. Red nodes and arrows indicate shared connections between rats and humans. AO anterior olfactory tubercle, Pyr pyriform cortex, aINS anterior insula, pINS posterior insula cortex, PrL prelimbic cortex, IL infralimbic cortex, OFC orbitofrontal cortex, ACC1

dorsal anterior cingulate cortex, ACC2 ventral anterior cingulate cortex, Cl claustrum, Sept septal nuclei, BNST bed nucleus of the stria terminalis, NAc nucleus accumbens core, NaSh nucleus accumbens shell, M1 primary motor cortex, S1 primary somatosensory cortex, PVT paraventricular thalamus, PVH paraventricular hypothalamus, CeA central nucleus of the amygdala, MeA medial nucleus of the amygdala, BLA basolateral nucleus of the amygdala, lPAG lateral periaqueductal gray, vlPAG ventrolateral periaqueductal gray, DR dorsal raphe, LC locus coeruleus, mPB medial parabrachial nucleus, lPB lateral parabrachial nucleus, RVM rostral ventromedial medulla, SRD subnucleus reticularis dorsalis, Gr gracile nucleus, ACC anterior cingulate cortex, MCC1 dorsal middle cingulate cortex, MCC2 ventral middle cingulate cortex, Cun cuneate nucleus.

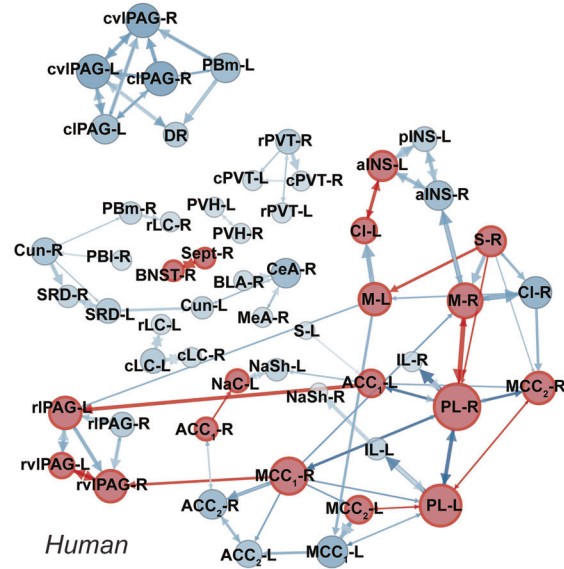

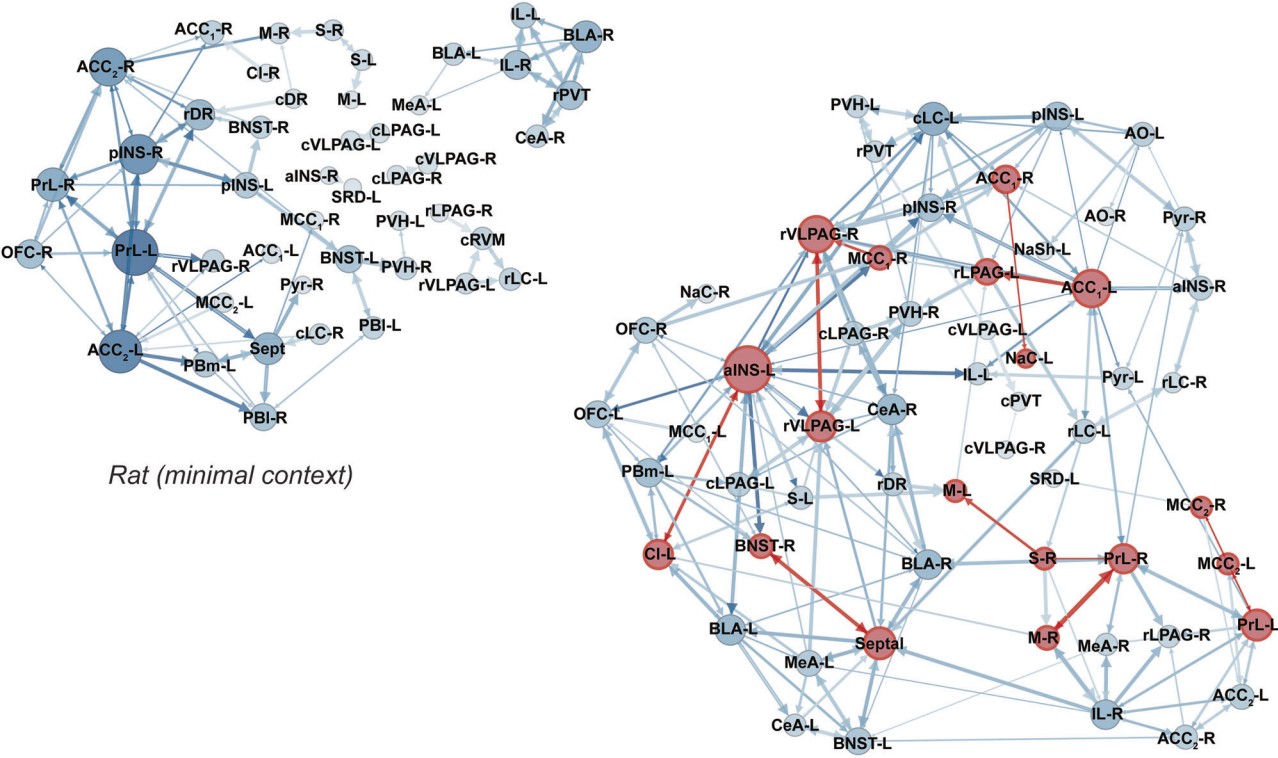

**Fig. 7 | Functional neural circuitry during nocebo hyperalgesia.** Functional connectivity maps between all ROIs were first generated using c-Fos expression in rats and beta-values in humans, with the top 100 strongest connections identified (Supplementary Fig. 1). The direction and strength of each of these connections was then applied based on known anatomical projections (Rat Connectome Project), or pruned if there was no evidence of a projection, and force directed graphs were created using Gephi. The size of the nodes reflect the degree of connectivity in the network. The size of the arrows indicate the strength of the projection. Red nodes and arrows indicate shared connections between rats and humans. AO anterior olfactory tubercle, Pyr pyriform cortex, aINS anterior insula, pINS posterior insula cortex, PrL prelimbic cortex, IL infralimbic cortex, OFC orbitofrontal cortex, ACC1 dorsal anterior cingulate cortex, ACC2 ventral anterior cingulate cortex, Cl claustrum, Sept septal nuclei, BNST bed nucleus of the stria terminalis, NAc nucleus accumbens core, NaSh, nucleus accumbens shell, M1 primary motor cortex, S1 primary somatosensory cortex, PVT paraventricular thalamus, PVH paraventricular hypothalamus, CeA central nucleus of the amygdala, MeA medial nucleus of the amygdala, BLA basolateral nucleus of the amygdala, lPAG lateral periaqueductal gray, vlPAG ventrolateral periaqueductal gray, DR dorsal raphe, LC locus coeruleus, mPB medial parabrachial nucleus, lPB lateral parabrachial nucleus, RVM rostral ventromedial medulla, SRD subnucleus reticularis dorsalis, Gr gracile nucleus, ACC anterior cingulate cortex, MCC1 dorsal middle cingulate cortex, MCC2 ventral middle cingulate cortex, Cun cuneate nucleus.

**Table 3 | Shared ROI-ROI functional and anatomical connections between humans and rats in the nocebo groups**

| Functional Connection | Pearson's r | | | Anatomical connection | Strength(s) |
|---|---|---|---|---|---|
| | Human | Rat (min) | Rat (enh) | | |
| PrL(R)-S1(R) | 0.783 | 0.771 | 0.991 | PrL(R)←S1(R) | 1 |
| PrL(R)-M1(R) | 0.769 | 0.790 | 0.940 | PrL(R)←→M1(R) | 4, 2 |
| PrL(R)-ACC-1(L) | 0.711 | 0.703 | 0.879 | PrL(R)←→ACC-1(L) | 2, 2 |
| S1(R)-PrL(L) | 0.775 | 0.771 | 0.952 | - | |
| S1(R)-M1(L) | 0.712 | 0.810 | 0.921 | S1(R)→M1(L) | 2 |
| M1(R)-M1(L) | 0.818 | 0.912 | 0.981 | - | |
| ACC-1(R)-NAc(L) | 0.699 | 0.770 | 0.890 | ACC-1(R)→NAc(L) | 1 |
| MCC-1(R)-rVLPAG(R) | 0.699 | 0.568 | 0.904 | MCC-1(R)→rVLPAG(R) | 2 |
| MCC-2(R)-PrL(L) | 0.704 | 0.676 | 0.932 | MCC-2(R)←→PrL(L) | 0.5, 1 |
| BNST(R)-Sept(R) | 0.706 | 0.734 | 0.880 | BNST(R)←→Sept(R) | 3, 3 |
| BLA(R)-CeA(R) | 0.711 | 0.941 | 0.916 | BLA(R)←→CeA(R) | 1, 3 |
| rVLPAG(R)-rVLPAG(L) | 0.948 | 0.876 | 0.908 | rVLPAG(R)←→rVLPAG(L) | 3, 3 |
| cVLPAG(R)-cLPAG(R) | 0.936 | 0.896 | 0.763 | cVLPAG(R)←→cLPAG(R) | 3, 0.5 |
| cLPAG(L)-cVLPAG(L) | 0.845 | 0.945 | 0.768 | cLPAG(L)←→cVLPAG(L) | 3, 3 |
| rLPAG(L)-M1(L) | 0.746 | 0.211 | 0.888 | rLPAG(L)←M1(L) | 1 |
| MCC-2(L)-M1(L) | 0.752 | 0.640 | 0.888 | - | |
| MCC-2(L)-PrL(L) | 0.728 | 0.823 | 0.916 | MCC-2(L)←→PrL(L) | 1, 1 |
| M1(L)-aINS(L) | 0.768 | 0.058 | 0.923 | - | |
| aINS(L)-Claus(L) | 0.828 | 0.419 | 0.967 | aINS(L)←→Claus(L) | 2.5, 2 |

Arrows indicate the direction(s) of the anatomical connections.

means that it represents a cumulative cellular response over the preceding 90–120 min from perfusion and fixation. Further, the reliance on post-mortem tissue for c-Fos quantification precludes a within-subjects design as was used in humans. Interestingly, both methodologies share the notable limitation of being unable to distinguish whether the observed changes in signal arise from excitatory or inhibitory neurons. However, ultimately, both techniques serve as proxies to capture population-level neural activation, with each signal presumed to reflect increased neuronal firing of action potentials, albeit with different spatiotemporal resolution. Future studies could potentially help to bridge these methodological differences by using fMRI in rats. However, despite significant advancements in small animal MRI imaging techniques, including the possibility of freely behaving fMRI and PET scanning, challenges such as motion artifacts, limited spatial resolution, and the technical and financial constraints of high-field imaging remain key obstacles to widespread implementation.

At the analysis level, we chose to use fMRI beta values, which collapses the signal across the scan into a single value for each region, thus providing the closest approximation to the cumulative nature of c-Fos expression. This approach also ensured that the functional connectivity maps derived from fMRI aligned more closely with the static connectivity patterns inferred from c-Fos densities. We also carefully selected and defined our brain ROIs to ensure that they were homologous across species, relying on known histological, functional, and anatomical similarities. Only 4 out of the 70 ROIs from each species lacked a clear equivalent in the other. In humans, the dlPFC has no discrete homologue in rodents[37–40]. Conversely, in rodents, the anterior olfactory nucleus and piriform cortex did not have a clear human counterpart, given the fundamental differences in olfactory processing across species[41–43]. We included these regions due to their known or putative involvement in conditioning and placebo responses in each species. Furthermore, by focusing on the top 20 ROIs with the largest changes in activity and the top 100 functional connections (or the relative differences from controls) in each species, we aimed to capture the most relevant neural changes during placebo and nocebo responses, while also accommodating for species-specific architecture.

Another important consideration at the analysis level given our statistical approach is how sample size might affect our results. This is particularly relevant for the functional connectivity analysis, given that it relies on a large number of pairwise correlations between ROIs. With fewer data points, the correlation coefficients (r values) become more susceptible to random noise, reducing their robustness and reliability, which increases the likelihood of spurious correlations or the failure to detect true underlying relationships. As such, we used 8 rats per group, as Terstege et al.[19] concluded that smaller group sizes are underpowered for c-Fos functional connectivity analyses. Similarly, the human sample sizes in our study were more than sufficient to provide reliable function connectivity estimates.

Finally, the pruning of the functional connectivity maps to retain only anatomically plausible connections was a key step in our analysis. This process identified 58 directional connections for placebo and 25 for nocebo (with variable strengths) that were shared by humans and rats, thus narrowing the focus to conserved pathways most likely involved in these responses. It is important to note that unlike the approach taken by many recent studies that focus on a single critical circuit, our aim was to provide a foundation for future research by identifying a small collection of specific, biologically plausible connections for functional interrogation. As such, these findings offer a starting point for defining the constellation of neural circuits necessary and sufficient for placebo and nocebo responses. It is our hope that our study will therefore act as a springboard for advancing translational efforts and refining our mechanistic understanding of these phenomena.

## Conclusions

This study offers a novel framework to investigate cross-species comparisons of the brain regions and neural circuits underlying placebo analgesia and nocebo hyperalgesia, revealing both conserved and species-specific patterns. By integrating functional connectivity with anatomical pruning, we identified a number of shared circuits likely responsible for placebo and nocebo responses across both rats and humans. Our findings underscore the translational relevance of rodent models for studying human pain processes

and targeting these shared neural circuits may hold promise for developing new therapies to treat pain conditions.

## Methods

### Overall experimental design

This study sought to directly compare the neural activity and circuits underlying placebo analgesia and nocebo hyperalgesia in both rats and humans, with the goal of identifying conserved neural circuits that could be targeted in the development of treatments for pain modulation. Rats provide a highly controlled model for dissecting specific neural mechanisms, while human data allows us to establish the relevance of these findings in a clinical context. To achieve this, we first quantified neural activity across 70 ROIs in the brain using c-Fos expression in rats and fMRI beta values in humans. Once neural activity was measured for each ROI, estimation statistics were applied to compare overall activity levels between control and placebo and nocebo conditions, identifying the regions with the most significant changes. This allowed for a direct comparison of neural activity patterns between species. Next, to define the neural circuits driving these responses, we examined functional connectivity—specifically, connections that were highly active during placebo and nocebo conditions. These functional connections were then pruned and weighted using known anatomical pathways from the Rat Connectome Project[20], ensuring that the resulting networks reflected biologically plausible circuits. No animal data was excluded from any stage of analysis. All graphs were created in either GraphPad Prism v9.3.1 or Gephi software 0.10.1 (Gephi Consortium, https://gephi.org/) and imported into Adobe Illustrator 2021 (v25.2) to create the figurework. This study was not preregistered.

### Rat experiments

**Animals and housing.** The experimental protocols used in this study were approved by the University of Sydney Animal Care and Ethics Committee (Project Number 1165). All procedures followed the guidelines outlined by the NHMRC's 'Code for the Care and Use of Animals in Research' and the NSW Animal Research Act (2007). The principles of the three R's (replacement, reduction, and refinement) were strictly applied to minimize pain and discomfort, and the study adhered to the IASP's 'Ethical Guidelines for Investigations of Experimental Pain in Conscious Animals.' We have complied with all relevant ethical regulations for animal use. Six-week-old male ($n = 50$) Sprague-Dawley rats (ARC, Perth, WA, Australia), weighing 170–220 g upon arrival were used for these experiments. Rats were housed in groups of four in individually ventilated cages, with *ad libitum* access to standard chow and water. Both the housing and testing rooms were kept at a controlled temperature ($22 \pm 1$ °C) and humidity (40–70%). To align with the rats' active period, the housing room operated on a reversed 12 h light/dark cycle, with lights off at 07:00. To control for potential experimenter-induced effects, the same experimenter (DB) conducted all handling, husbandry, and procedures across all rodent groups, ensuring consistency in interactions and minimizing variability in conditioning and testing.

**Eliciting placebo analgesia and nocebo hyperalgesia.** The experimental design, behavioural procedures and behavioural results have been reported previously[4]. A response conditioning protocol was used to induce placebo analgesia and nocebo hyperalgesia in rats with a chronic constriction injury (CCI) of the sciatic nerve. Six days post-injury, rats underwent conditioning using a hot/cold plate analgesiometer, with 10 trials conducted over 5 consecutive days (5 morning and 5 afternoon sessions). Placebo groups were conditioned at a non-noxious thermo-neutral temperature (30 °C), while nocebo groups were conditioned at a noxious cold temperature (4 °C). On Test Day, both groups were exposed to a mildly noxious cold stimulus (20 °C), and differences in hind paw withdrawal responses were measured to assess conditioned placebo or nocebo effects by comparing them to a natural history control group, which were conditioned and tested at 20 °C. To assess the impact of contextual salience on placebo and nocebo effects, enhanced context groups were conditioned with additional sensory cues. Rats were randomly assigned to each group, and all data, including video recordings, are available upon request.

All response-conditioning sessions took place in a temperature- and humidity-controlled room ($21 \pm 1$°C, 40–75% humidity) separate from the housing room. The control group and the minimal context groups were tested in a context designed to minimize the strength and saliency of sensory cues, with rats tested under dim red lighting ($\sim 10$ lux) in a quiet, odour-free environment. Enhanced context groups were conditioned in a custom-built chamber (60 cm x 45 cm x 30 cm) under medium-level white lighting ($\sim 50$ lux), wideband white noise (75–80 dB), and a strong vanilla scent from a plug-in diffuser (Fig. 1o). The chamber was cleaned with an apple-scented disinfecting wipe between tests to maintain olfactory consistency. To prevent the possibility of lingering odours from affecting the minimal context groups, enhanced context groups were always tested at the end of the testing period. Rats were tested 10 times over 5 consecutive days, with 5 morning and 5 afternoon sessions. Testing began approximately +2 h from lights off for morning sessions and approximately +8 h from lights off for afternoon sessions, with each cage of rats tested in the same sequence at ~5 min intervals. Perfusions were performed 90 minutes after testing under deep pentobarbital anaesthesia, using 400 ml of cold heparinised saline followed by 400 ml of cold 4% paraformaldehyde in sodium acetate-borate buffer, pH 9.6.

**Chronic constriction injury (CCI) surgery.** All rats underwent a unilateral chronic constriction injury (CCI) of the sciatic nerve, following the method initially detailed by Bennett and Xie (1988)[44]. At the time of surgery, the rats weighed between 240 and 280 grams. The procedure began with anaesthesia induced and maintained using isoflurane (5% for induction and 2.5% for maintenance) delivered in 100% oxygen at a flow rate of 1.5 L/min. After achieving a surgical plane of anaesthesia, the right hind limb was shaved and sterilized with povidone-iodine. A 2 cm incision was made parallel to the femur, and blunt dissection of the biceps femoris was performed to expose the sciatic nerve. Four chrome catgut ligatures (5-0, Johnson & Johnson) were loosely tied around the nerve, spaced 1 mm apart, just proximal to its trifurcation. Care was taken to ensure that the ligatures compressed the nerve without obstructing epineural blood flow. The nerve was then repositioned, and the incision was closed with Michel clips. A few drops of lignocaine (20 mg/ml) were applied to the incision site, followed by a dusting of topical antibiotic powder (Tricin®, Jurox). To prevent licking of the wound, a mixture of petroleum jelly and quinine was applied. Post-surgery, the rats were placed in individual cages and allowed to recover until they regained mobility and alertness (approximately 30 min) before being returned to their home cages. Animal welfare was monitored daily post-surgery and signs of autotomy and standard criteria were used for humane endpoints, but no animal met these euthanasia criteria during the study.

**c-Fos Immunohistochemistry.** The brains of 8 rats per group were selected and sectioned at 50 µm using a cryostat (Leica CM1950) into a 1 in 8 series. One series from each brain was stained for c-Fos. Free-floating sections were first washed in 0.1 M PBS (3 × 10 min) at room temperature. Sections were then permeabilized in 50% ethanol for 30 min, followed by quenching of endogenous peroxidase activity with 3% hydrogen peroxide in 50% ethanol for 30 min. After further washes (3 × 10 minutes in PBS), sections were incubated in 10% normal horse serum (NHS) in 0.1 M PBS for 30 minutes to block non-specific binding. Sections were then incubated overnight at 4 °C with polyclonal rabbit anti-c-Fos IgG (1:3000 in 2% NHS/PBS; Abcam, RRID:AB_2737414). The following day, sections were washed in PBS and incubated with horse anti-rabbit IgG (1:500 in 2% NHS/PBS; Vector Laboratories, RRID:AB_2336201) for 2 hours at room temperature. After additional washes, sections were incubated in ExtrAvidin Peroxidase (1:1000 in PBS; Sigma-Aldrich) for 2.5 h. c-Fos immunoreactivity was visualized using 3,3'-diaminobenzidine tetrahydrochloride (DAB) as the chromogen, with the glucose

oxidase method, producing a brown precipitate. The reaction was carried out on ice and stopped after ~15 min with PBS washes (4 × 10 min), once optimal contrast was achieved. Sections were mounted on gelatin-coated slides, allowed to dry for 48 h, dehydrated through ascending ethanol series, cleared in histolene, and coverslipped with DPX mounting medium.

The brain ROIs were carefully selected based on their involvement in key processes such as pain transmission and modulation, as well as the processing of contextual information, including olfactory cues. A number of regions, such as orbitofrontal cortex, NAc and claustrum were also included due to their critical roles in integrating sensory information, motivational states, and decision-making processes—elements central to both placebo and nocebo responses. In total, 70 brain regions were analyzed to capture the full scope of these mechanisms. All images were captured using a Gryphax Kapella camera (Jenoptik, Jena, Germany). ROIs were identified based on the cytoarchitectural features of the tissue, with the stereotaxic rat atlas of Paxinos and Watson (2005) serving as a guide. Each microscope slide was coded to blind the experimenter (DB) during both image capture and analysis. For each ROI, 1, 2, or 3 slices per brain were analyzed and averaged, depending on the region. ROIs were manually outlined using ImageJ software, and c-Fos density was calculated for each ROI in each brain.

### Human experiments

**Participants and apparatus.** All experimental procedures were approved through the University of Sydney Human Research Ethics Committee (HREC:2019/037), consistent with the Declaration of Helsinki. All ethical regulations relevant to human research participants were followed. 46 healthy control participants were recruited for this study using recruitment flyers and word of mouth. All 46 participants underwent the placebo component, and 25 of these participants also underwent the nocebo component. All participants provided written informed consent on arrival. Inclusion criteria for participants were as follows: (i) participants must be over the age of 18, (ii) understand English and (iii) be able to provide written and verbal informed consent. The exclusion criteria were as follows: (i) not be currently or possibly pregnant, (ii) not have any metallic implants and (iii) not currently experiencing pain or have been diagnosed with any chronic pain condition. Participants were provided with an emergency buzzer whilst inside the scanner so that they could stop the experiment at any time. Before exiting the study, participants were informed as to the necessary deception and true methodology of the experiment both verbally and through a written statement.

Noxious thermal stimuli were delivered throughout the protocol using a Peltier-element thermode device (Medoc LTD Advanced Medical Systems, Rimat Yishai, Israel), connected to a thermal sensory analyser which delivered eight stimuli at a pre-programmed temperature for 15 s (2.5 s ramp up, 10 s plateau, 2.5 s ramp down), each separated by a 15 s inter-stimulus interval at a baseline temperature of 32 degrees Celsius (°C). Functional magnetic resonance imaging (fMRI) sequences were acquired using a whole-body Siemens MAGNETOM 7 Tesla (7 T) MRI system (Siemens Healthcare, Erlangen, Germany) with a combined single-channel transmit and 32-channel receive head coil (Nova Medical, Wilmington MA, USA). located at the Melbourne Brain Centre Imaging Unit in Melbourne, Victoria.

**Placebo and nocebo conditioning.** On entering the study, participants were shown three creams: a control cream described as Vaseline, a placebo cream labelled and described to contain the analgesic additive Lidocaine, and a nocebo cream labelled and described to contain the topical irritant, Capsaicin. The three creams were placed in adjacent locations on participants' right forearm and remained applied for five minutes to "take effect". During this time, a thermal calibration protocol was conducted, with the thermode attached to participants' left forearm, delivering a randomized series of temperatures between 44 and 48.5 °C in the timings described above. Participants were informed this protocol

was conducted to establish their moderate pain intensity – a temperature eliciting between 4 to 5 out of 10 on a 10-point scale (0 = no pain; 10 = worst pain imaginable), and that this moderate intensity would be used for the remainder of the study applied to each of the three creams. In reality, three temperatures were recorded, one eliciting a low pain (2-3/10 VAS), one eliciting a high pain (6-7/10 VAS), as well as the initially described moderate intensity.

The three creams were then removed from participants' forearms with the experimenter wearing gloves to enhance belief the creams contained active ingredients, and a response-conditioning protocol was conducted. Different intensity noxious stimuli were applied to each of the three cream sites (low intensity to placebo, moderate intensity to control, high intensity to nocebo), despite informing participants that all three cream sites would be receiving identical stimuli (i.e., moderate intensity). Participants recorded their pain in real time using a visual analogue scale, building belief that the respective placebo and nocebo creams were taking effect relative to the control cream. The order of stimulation and locations of the placebo and nocebo creams proximal or distal relative to control was counterbalanced between participants to reduce ordering or sensitivity effects. This concluded day 1. All participants returned the following day at the same time for a second round of response conditioning (reinforcement) and subsequent fMRI scanning (Fig. 1a–d). This reinforcement protocol took place immediately before the fMRI scan and on the left forearm, in which the creams and noxious stimuli were again used at the same individually calibrated low, moderate and high intensity on day 1. This was conducted to strengthen the conditioned effects and confirm the temperature for the moderate intensity stimulus for fMRI testing for each participant.

**Placebo and Nocebo Test.** Five minutes before entering the MRI scanner, the three creams were applied to the right forearm. Over the course of four independent fMRI series, each of the cream sites were stimulated a total of eight times in the timings described above (15-second stimulus, 15-second inter-stimulus interval). Unlike in the response conditioning sessions, during fMRI testing all three cream sites received identical intensity moderate stimuli, with the mean difference between VAS scores recorded during control-site stimulation relative to the placebo-site encoding their placebo analgesia response, and vice versa for the nocebo-site and their nocebo hyperalgesia response. The control site was stimulated twice to provide a "pre" and "post" measurement for both placebo analgesia and nocebo hyperalgesia, and this order was kept counterbalanced in the same order as during conditioning (i.e., 50% of participants received control- and placebo-site series first and vice versa). During this test phase, participants recorded their ongoing pain in real time during a digital and MR-compatible VAS, reflected on a screen above with the position of a slider controlled using a two-button button box.

**Determining placebo and nocebo responders.** Using test phase VAS ratings, participants' mean pain to each of the eight stimuli delivered to either the control- and placebo cream sites, or the control- and nocebo cream sites were entered into a bootstrapped permutation model, where 10,000 artificial samples were generated, sampled with replacement, encoding both their mean control and placebo, or control and nocebo responses. These responses were significance tested with a significant reduction of pain during placebo relative to control, or significant enhancement of pain during nocebo relative to control indicating a placebo or nocebo responder, respectively. From our sample, 22/46 participants were identified as placebo responders (46%), and 14/25 participants identified as nocebo responders (56%). Neither age nor sex significantly influenced the likelihood of being classified as a responder in either condition (see Supplementary Table 1). Whilst fMRI series were collected on the entire sample and are published elsewhere[13,14], this investigation involves analyses conducted solely on these two responder cohorts. Additionally, given that it has previously been shown that participant-experimenter interactions (including gender) can influence

placebo and nocebo responses, the same experimenter (LC) and radiographer carried out all procedures for all participants.

**MRI acquisition and preprocessing parameters.** All image series were acquired with participants positioned supine within the head coil, with sponges inserted to support the head and minimise lateral and translational movement. A T1-weighted anatomical image set covering the whole brain was collected (repetition time=5000 ms, echo time=3.1 ms, raw voxel size=0.73 × 0.73 × 0.73 mm, 224 sagittal slices, scan time=7 mins). Each of the four fMRI acquisitions consisted of 134 gradient echo echo-planar measurements using BOLD contrast covering the entire brain. Images were acquired interleaved with a multi-band factor of four and an acceleration factor of three (repetition time=2500 ms, echo time=26 ms; raw voxel size=1.0 × 1.0 × 1.2 mm, 124 axial slices, scan time=6:25 mins).

Image preprocessing and statistical analyses were performed using SPM12 (Friston, 2003) and custom software[45]. The first five volumes of each scan were removed from the model due to excessive signal saturation from the scanner. The remaining 129 functional images were slice-time and motion corrected and the resulting 6 directional movement parameters were inspected to ensure that all fMRI scans had no greater than 1 mm of linear movement or 0.5 degrees of rotation movement in any direction. Images were then linearly detrended to remove global signal changes, physiological noise relating to cardiac and respiratory frequency was removed using the DRIFTER toolbox[46], and the 6-parameter movement related signal changes were modelled and removed using a linear modelling of realignment parameters procedure[47]. From this point, two image sets were created: one wholebrain and one brainstem-isolated functional image set.

For wholebrain images, each individual's fMRI image sets were resliced to 1 mm isotropic voxels and co-registered to their own T1-weighted anatomical. The T1 was then spatially normalized to the MNI152 template in Montreal Neurological Institute (MNI) space using the computational anatomy toolbox (CAT)[48], and these parameters applied to the fMRI image sets. The normalized fMRI images were then spatially smoothed using a 6 mm full-width at half maximum Gaussian filter. For brainstem-isolated images, the Spatially Unbiased Infratentorial Template (SUIT) toolbox was used to first isolate in the T1 image series a section of the image encompassing the brainstem and cerebellum. Manual made masks were then generated covering the rostrocaudal extent of the brainstem and cerebellum in both the brainstem isolated T1 and wholebrain fMRI image series, resliced into 0.5 mm isotropic voxels. Both image series were cropped using the manual masks, and the T1 normalized to the SUIT template in MNI space using the SUIT normalize function, with these parameters applied to the cropped fMRI series. The normalized fMRI images were then spatially smoothed using a 1 mm FWHM gaussian filter. In both the wholebrain and brainstem-isolated image series, signal intensity changes were determined by applying a repeating boxcar model encoding scan volumes where noxious stimuli were present was applied, convolved with a canonical hemodynamic response function. This analysis resulted in brain maps in which each voxels value represented the magnitude of signal intensity changes during each noxious stimulus period.

**VOI generation and beta-value extraction.** Individual VOIs were generated for each cortical and brainstem pain-responsive site, derived from the Human Connectome Project atlas extended (HCPex)[49] for the cortex and subcortex, and hand drawn with reference to Mai and Paxinos' Atlas of the Human Brain[50]. In total, 71 VOIs were generated and are listed in Supplementary Table 2. A measurement of average signal intensity, that is, relative increases or decreases in neuronal activation during the application of noxious stimuli relative to baseline periods of the fMRI scan (i.e., the 90-second period preceding the first noxious stimulus, and every 15-second interval between noxious stimuli), were extracted from each of the matched control and placebo or control and

nocebo contrast images. These values were then recorded in a matrix table for further analyses.

## Statistics and Reproducibility
**Placebo and nocebo-induced changes in regional activity.** For the calculation of placebo and nocebo-related changes in overall regional activity, we used estimation statistics to quantify the differences between the placebo and control groups and the nocebo and control groups for all 70 ROIs. ROIs were then ranked based on the effect size (Cohen's d) and the top 20 regions with the largest effect sizes were identified for both placebo and nocebo conditions (Figs. 3, 4). We chose to select specifically 20 out of the 70 regions investigated as this is roughly the number of regions across the brain that have shown consistent involvement in placebo and nocebo effects based on previous key studies and meta-analyses[7,11,13,14,21,29]. Estimation statistics were computed using the web application built by Hung Nguyen (estimationstats.com), which utilizes the Python code developed by Ho et al. (2019)[51]. For each pairwise test, the Cohen's d and their associated 95% confidence intervals were calculated using bias-corrected and accelerated bootstrap resampling with replacement, with 5000 bootstrap samples applied per test.

We opted for estimation statistics over classical significance testing due to their ability to offer richer, more nuanced insights into the magnitude and precision of the observed changes in neural activity. By focusing on effect sizes, this method captures the magnitude and uncertainty of neural activity changes, highlighting the most relevant patterns without being restricted by arbitrary significance thresholds. This approach was crucial for comparing species-specific and shared patterns of neural connectivity, ensuring that we prioritized meaningful biological differences over potentially spurious statistical results.

**c-Fos and beta-value functional connectivity and neural circuitry.** For both the c-Fos density and the beta-values, a correlation matrix was generated for each group (control, placebo, and nocebo) using GraphPad Prism 9. Each matrix represented the pairwise correlations between the 70 ROIs, capturing the strength of connectivity based on Pearson's r-values. This yielded 2415 unique correlations between the 70 ROIs for each group. For the control groups, we then isolated the top 100 largest r-values in each species to create an adjacency matrix (created in Microsoft Excel), representing the most robust functional connections. The selection of 100 functional connections was based on the need to balance interpretability and comprehensiveness. This number provides a sufficiently broad representation of the network structure, allowing us to capture key connectivity patterns without overwhelming the analysis with an excessive number of connections. A smaller subset (e.g., 50 connections) risked omitting important interactions critical for understanding placebo and nocebo effects, while a larger subset (e.g., 500 connections) or complete set would introduce noise and reduce clarity in identifying meaningful network differences. Selecting the top 100 connections ensured that we could effectively characterize large-scale network alterations while maintaining a focused and interpretable dataset, ultimately facilitating meaningful cross-species comparisons.

These 100 connections were visualized in network plots for both rat and human control groups (Supplementary Fig. 1). To maintain consistency, the same r-value cut-off that isolated the top 100 connections in the control groups was applied to the placebo and nocebo groups. This allowed us to determine changes in overall connectivity during placebo analgesia and nocebo hyperalgesia, and the connectivity patterns were also plotted for each group. We then identified shared connections between the rat and human groups (Tables 1–3) to highlight species-conserved connections.

To visualize the functional connectivity graphs, we used Gephi software v0.10.1. The adjacency matrices, which were calculated from the correlation matrices, were imported into Gephi as undirected connections. We utilized the circular layout, with nodes arranged by their ID. The

**Article**

darkness of each node was scaled according to the number of connections it held, allowing for easy identification of hub regions. After generating the graphs, they were exported into Adobe Illustrator for further refinement. Shared connections between rats and humans were manually highlighted in red to visualize conserved pathways across species.

Finally, to give us the functional connectivity circuitry of each group we pruned these networks by cross-referencing with anatomical data from the Rat Connectome Project (https://neuroviisas.med.uni-rostock.de/connectome/index.php), which catalogues the existence and strength of anatomical connections between brain regions based on neuroanatomical tract tracing studies. This approach ensured that only connections with known anatomical pathways remained in the analysis. All others were discarded. The connectome provides further details about the strength and direction of each connection, assigning connection strengths ranging from 0.5 (very light) to 4 (very strong), forming the basis of the final neural circuitry model. This pruning step allowed us to refine the functional networks into anatomically grounded circuits, providing insights into the neural circuits underlying placebo analgesia and nocebo hyperalgesia across species. Similar to the functional connectivity analysis, adjacency matrices were generated from the correlation matrices and imported into Gephi as directed connections.

We applied the Force Atlas layout with the following parameters: inertia = 0.1, repulsion strength = 1000, attraction strength = 10, maximum displacement = 1000, auto stabilize function = on, autostab strength = 80, autostab sensibility = 0.2, gravity = 3000, attraction distribution = off, adjust by sizes = on, and speed = 1.0. Node size was determined by the degree of connectivity, where more connected nodes appeared larger, while the size of the arrows represented the strength of the anatomical projection. This method allowed us to generate directed graphs that visually emphasize the most strongly connected nodes and circuits, reflecting the overall structure of the neural networks across species.

### Reporting summary

Further information on research design is available in the Nature Portfolio Reporting Summary linked to this article.

### Data availability

All source data for all graphs and Figures in the paper can be found in the Supplementary Data. All other data are available from the corresponding author upon reasonable request. This includes raw and processed imaging data, behavioural datasets, and analysis scripts used in this study. Due to ethical considerations, raw human data have been de-identified to ensure privacy and confidentiality in compliance with applicable guidelines and regulations. Access to these data may require approval from the appropriate ethics board and completion of a data-sharing agreement. Animal data, including c-Fos expression maps and functional connectivity matrices, as well as the custom analysis code, are available directly from the corresponding author upon request.

### Code availability

MRI Image preprocessing and statistical analyses were performed using: (i) Statistical Parametric Mapping (SPM12) and (ii) custom software (Diedrichsen, J (2006), 'A spatially unbiased atlas template of the human cerebellum', Neuroimage, vol. 33, no. 1, pp. 127-38.) and (Macey, PM, Macey, KE, Kumar, R & Harper, RM (2004), 'A method for removal of global effects from fMRI time series', Neuroimage, vol. 22, no. 1, pp. 360-6.), (iii) the DRIFTER toolbox (Sarkka, S, Solin, A, Nummenmaa, A, Vehtari, A, Auranen, T, Vanni, S & Lin, FH (2012), 'Dynamic retrospective filtering of physiological noise in BOLD fMRI: DRIFTER', Neuroimage, vol. 60, no. 2, pp. 1517-27), and (iv) CAT (Gaser, C, Dahnke, R, Thompson, PM, Kurth, F, Luders, E & The Alzheimer's Disease Neuroimaging, I (2024), 'CAT: a computational anatomy toolbox for the analysis of structural MRI data', Gigascience, vol. 13.). For both the c-Fos density and the beta-values, a correlation matrix was generated for each group using GraphPad Prism 9.3.1. Adjacency matrices were created using Microsoft Excel. For graphing

of networks, Gephi Software v0.10.1 was used. Specific parameters are described in the manuscript. Estimation statistics were computed using the web application built by Hung Nguyen (estimationstats.com), which utilizes the Python code developed by (Ho, J, Tumkaya, T, Aryal, S, Choi, H & Claridge-Chang, A (2019), 'Moving beyond P values: data analysis with estimation graphics', Nat Methods, vol. 16, no. 7, pp. 565-566.).

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

## Acknowledgements

We wish to thank the many volunteers in this study. The authors acknowledge the facilities and scientific and technical assistance of the National Imaging Facility, a National Collaborative Research Infrastructure Strategy (NCRIS) capability, at Monash University. This work was funded by the National Health and Medical Research Council of Australia Grant 1130280 and the NWG Macintosh Memorial Grant, University of Sydney, Australia.

## Author contributions

Conceptualization: D.B., L.C., L.H., K.K.; Methodology: D.B., L.C., L.H., K.K.; Software: D.B., L.C., L.H.; Validation: D.B., L.C., L.H., K.K.; Formal analysis: D.B., L.C.; Investigation: D.B., L.C.; Resources: L.H., K.K.; Data curation: D.B., L.C.; Writing – original draft: D.B., L.C., L.H., K.K.; Writing – Reviewing & editing: D.B., L.C., L.H., K.K.; Visualization: D.B. Supervision: L.H., K.K.; Project administration: D.B., L.C., L.H., K.K. Funding acquisition: L.H., K.K.

## Competing interests

The authors declare that they have no known competing financial interests or personal relationships that could have appeared to influence the work reported in this paper.
