## [Transparent Peer Review file · Communications Biology]

Direct comparisons of neural activity during placebo analgesia and nocebo hyperalgesia between humans and rats

Corresponding Author: Dr Damien Boorman

Version 0:

Reviewer comments:

Reviewer #1

(Remarks to the Author)

This study represents a significant advancement in the field of placebo and nocebo research, offering a novel framework for cross-species comparisons of neural circuits. The findings are robust and well-supported, with important implications for translational research. Addressing the identified weaknesses would further strengthen the manuscript and enhance its impact.

Weakness:

1. The discussion of methodological differences between c-Fos and fMRI could be expanded, particularly regarding their implications for interpreting the findings.
2. The pruning criteria for functional connectivity maps are not fully detailed, which may limit reproducibility.
3. The potential impact of sample size and the reliance on post-mortem tissue for c-Fos analysis is not thoroughly addressed.

Recommendations:

1. Expand the discussion on the functional implications of the observed differences in network organization between placebo and nocebo conditions.
2. Provide additional details on the selection criteria for the top 20 ROIs and top 100 functional connections to enhance transparency.
3. Include a more detailed explanation of the pruning criteria for functional connectivity maps.
4. Consider adding a brief summary of the key findings from Tables 2 and 3 in the main text to improve accessibility.
5. Explore the potential impact of sample size and methodological limitations in greater depth.

Reviewer #2

(Remarks to the Author)

The authors aimed to investigate whether the neurobiological pathways underlying the placebo and nocebo effects are conserved across species, specifically between humans and rats. This is a relevant research question. The authors employed an innovative approach to address this, and the sample size appears to be appropriate. Several strengths of the study are worth highlighting, including the inclusion of the brainstem in the analysis and the integration of functional connectivity informed by underlying anatomical connections.

In this discussion, the authors addressed some of my concerns. For instance, while c-Fos immunohistochemistry and fMRI may currently be the most comparable methods available, they still differ in terms of temporal resolution.

Below are some aspects to clarify/add additional information:

Methods:

1. Human Sample: Please provide additional details regarding the inclusion and exclusion criteria for the study participants, as well as socio-demographic information about the subjects included in the analysis. Additionally, include a comparison between responders and non-responders to better understand any potential differences between these groups.
2. Human experiments: Regarding the timing of the sessions, were all sessions spaced one day apart? Please clarify this

and provide more details, including the average interval between session 1 and session 2.

3. VOI generation and beta-value extraction – The authors wrote ‘A measurement of average signal intensity, that is, relative increases or decreases in neuronal activation during the application of noxious stimuli relative to baseline periods of the fMRI scan,’ - what was considered ‘baseline’ in this case?

4. Placebo and nocebo test – Were the expectations/expectancies measured? How did they relate to the placebo/nocebo effect recorded?

5. Placebo and nocebo-induced changes in regional activity – While I agree that the use of estimation statistics provides a nuanced understanding of neural activity, I wonder whether alternative approaches, such as multivariate pattern analysis (e.g., brain signatures), could offer comparable information without requiring a predefined selection of regions to focus on. Please clarify the rationale for selecting 20 regions out of the 70 studied.

6. Please clarify the definition of the term "control group" as used throughout the text

Discussion:

1. Please expand on the potential bias introduced by using rats with chronic constriction injury of the sciatic nerve compared to healthy human participants. How might this difference affect the generalizability of the findings?

2. One of the key questions in the field of placebo effect research is the exploration of interindividual differences. While this study did not aim to address this issue, it would be valuable to consider how the findings might contribute to our understanding of these variations in future research.

3. Additionally, while the impact of verbal suggestions cannot be directly studied, the interactions between both humans and rats with the experimenter may influence the results. Was this potential confounding factor accounted for in the study design? If so, please provide details on how it was managed.

Version 1:

Reviewer comments:

Reviewer #1

(Remarks to the Author)

The author's response is comprehensive and robust, addressing the reviewer's key concerns (e.g., methodological details, statistical power, study limitations) through detailed revisions and thoughtful discussions. The manuscript's transparency and scientific quality are significantly enhanced by these revisions. Although some responses could be more concise, and the discussion of future research and alternative methods more specific, the author has thoroughly resolved the reviewer's main concerns.

Overall, the response is appropriate and lays a solid foundation for the manuscript's acceptance.

Reviewer #2

(Remarks to the Author)

All my concerns and comments were addressed.

Damien Boorman, PhD
School of Medical Sciences (Neuroscience)
The Brain and Mind Centre
100 Mallett Street, Camperdown, NSW, Australia, 2006

15 February 2025

Dear reviewers,

Thank you for your insightful and helpful comments. We have addressed them to the best of our ability. Please find below our point-by-point reply to each comment and the relevant edits made to the manuscript.

Reviewer #1 Comments:

This study represents a significant advancement in the field of placebo and nocebo research, offering a novel framework for cross-species comparisons of neural circuits. The findings are robust and well-supported, with important implications for translational research. Addressing the identified weaknesses would further strengthen the manuscript and enhance its impact.

Weakness:

- 1. The discussion of methodological differences between c-Fos and fMRI could be expanded, particularly regarding their implications for interpreting the findings.*
- 2. The pruning criteria for functional connectivity maps are not fully detailed, which may limit reproducibility.*
- 3. The potential impact of sample size and the reliance on post-mortem tissue for c-Fos analysis is not thoroughly addressed.*

Recommendations:

- 1. Expand the discussion on the functional implications of the observed differences in network organization between placebo and nocebo conditions.*

The following paragraph has been added to the discussion:

“Finally, our findings reveal large and important differences in both the regional activity and the functional connectivity between placebo and nocebo conditions, with distinct networks emerging for each. The circuits recruited in placebo and nocebo show minimal overlap, suggesting they rely on fundamentally different neural mechanisms. In other words, placebo analgesia and nocebo hyperalgesia do not merely represent opposite shifts in activity within a shared pain circuit, but rather engage distinct and functionally separate neural pathways as previously described {Freeman, 2015 #76}. Additionally, the minimal overlap in functional connectivity between rats and humans in the nocebo condition suggests greater species-specific variability in how the nocebo response conditioning influences pain processing. This underscores the need for future research to investigate placebo and nocebo as fundamentally distinct phenomena to better understand their unique mechanisms and develop more precise therapeutic strategies.”

**Brain and Mind Centre
School of Medical Sciences
(Neuroscience)**

**Faculty of Medicine and Health
Rm 507, 94-100 Mallett Street, M02G
Camperdown
NSW 2050 Australia**

**T +61 2 9036 5314
M +61 425 222 749
E dboo2217@uni.sydney.edu.au**

ABN 15 211 513
464
CRICOS 00026A

2. Provide additional details on the selection criteria for the top 20 ROIs and top 100 functional connections to enhance transparency.

The selection of 20 regions to compare between species from the 70 studied was based on the approximate number of regions that show consistent involvement in placebo and nocebo effects from previous studies and meta-analyses. We have added the following to the methods:

“For the calculation of placebo and nocebo-related changes in overall regional activity, we used estimation statistics to quantify the differences between the placebo and control groups and the nocebo and control groups for all 70 ROIs. ROIs were then ranked based on the effect size (Cohen’s d) and the top 20 regions with the largest effect sizes were identified for both placebo and nocebo conditions (Figures 3, 4). We chose to select specifically 20 out of the 70 regions investigated as this is roughly the number of regions across the brain that have shown consistent involvement in placebo and nocebo effects based on previous key studies and meta-analyses (Tinnermann et al., 2017; Xu et al., 2018; Zeng et al., 2018; Crawford et al., 2021; Zunhammer et al., 2021; Crawford et al., 2023a).

In regard to the top 100 functional connections we have added the following to the methods:

“For the control groups, we isolated the top 100 largest r-values in each species to create an adjacency matrix (created in Microsoft Excel), representing the most robust functional connections. The selection of the top 100 functional connections was based on the need to balance interpretability and comprehensiveness. This number provides a sufficiently broad representation of the network structure, allowing us to capture key connectivity patterns without overwhelming the analysis with an excessive number of connections. A smaller subset (e.g., 50 connections) risked omitting important interactions critical for understanding placebo and nocebo effects, while a larger subset (e.g., 500 connections) or complete set would introduce noise and reduce clarity in identifying meaningful network differences. Selecting the top 100 connections ensured that we could effectively characterize large-scale network alterations while maintaining a focused and interpretable dataset, ultimately facilitating meaningful cross-species comparisons. These 100 connections were visualized in network plots for both rat and human control groups (Supplementary Figure 1).”

3. Include a more detailed explanation of the pruning criteria for functional connectivity maps.

We have now added a more detailed description for the pruning criteria in the methods, and provided a link to the Rat Connectome Project.

“Finally, we pruned these networks by cross-referencing with anatomical data from the Rat Connectome Project (<https://neuroviisas.med.uni-rostock.de/connectome/index.php>), which catalogues the existence and strength of anatomical connections between brain regions based on neuroanatomical tract tracing studies. This approach ensured that only connections with known anatomical pathways remained in the analysis. All others were discarded. The connectome provides further details about the strength and direction of each connection, assigning connection strengths ranging from 0.5 (very light) to 4 (very strong), forming the basis of the final neural circuitry model. This pruning step allowed us to refine the functional networks into anatomically grounded circuits, providing insights into the neural circuits underlying placebo analgesia and nocebo hyperalgesia across species. Similar to the functional connectivity analysis, adjacency matrices were generated from the correlation matrices and imported into Gephi as directed connections.”

4. Consider adding a brief summary of the key findings from Tables 2 and 3 in the main text to improve accessibility.

To improve accessibility, we have now clarified that Table 2 summarizes and details the data from Figure 6 and Table 3 summarizes and details the data from Figure 7. We have modified the text:

“Figure 6 shows the pruned placebo networks for humans and rats and all shared connections are detailed in Table 2, which includes their direction and the strength of these pathways. Several key hub regions...”

“Figure 7 shows the pruned nocebo networks for humans and rats and all shared connections for nocebo groups are detailed in Table 3, which includes their direction and the strength of these pathways. Key hub regions...”

5. Explore the potential impact of sample size and methodological limitations in greater depth.

In regard to impact of sample size we have added this paragraph to the discussion:

“Another important consideration at the analysis level given our statistical approach is how sample size might affect our results. This is particularly relevant for the functional connectivity analysis, given that it relies on a large number of pairwise correlations between ROIs. With fewer data points, the correlation coefficients (r values) become more susceptible to random noise, reducing their robustness and reliability, which increases the likelihood of spurious correlations or the failure to detect true underlying relationships. As such, we used 8 rats per group as Terstege et al. (2022) concluded that smaller group sizes are underpowered for c-Fos functional connectivity analyses. Similarly, the human sample sizes in our study were more than sufficient to provide reliable function connectivity estimates.”

In regard to other methodological limitations, we have added this paragraph to the discussion:

“One notable difference to discuss between the human and rodent models is that the rats were given a CCI while the human participants were healthy and uninjured. Importantly, for our purposes, both human and rodent paradigms used in our study relied on acute stimulus-evoked responses rather than spontaneous pain, making them comparable in assessing placebo- and nocebo- induced pain modulation. However, it remains possible that the CCI model alters neural processing in response to pain, placebo, and nocebo, which could introduce differences in activation patterns compared to healthy individuals. While there has yet to be an animal study that investigates whether this is the case, (Wang et al., 2022) demonstrated that participants with chronic neuropathic pain (TMD) exhibit comparable levels of placebo analgesia to healthy individuals, indicating that the neural circuitry underlying acute placebo responses remains intact in people despite the presence of chronic pain. This suggests that while chronic pain conditions may influence certain aspects of pain processing, they do not necessarily impair the fundamental mechanisms driving placebo effects. Given these considerations, it would be worthwhile to replicate our findings using uninjured rodent models, particularly since recent studies have developed more directly comparable models of placebo analgesia and nocebo hyperalgesia in healthy mice {Kimmey, 2025 #79; Poulson, 2025 #80}. Expanding research to include both injured and uninjured models will help clarify the extent to which chronic pain state influences placebo and nocebo mechanisms and enhance the generalizability of these findings.”

Reviewer #2 Comments:

The authors aimed to investigate whether the neurobiological pathways underlying the placebo and nocebo effects are conserved across species, specifically between humans and rats. This is a relevant research question. The authors employed an innovative approach to address this, and the sample size appears to be appropriate. Several strengths of the study are worth highlighting, including the inclusion of the brainstem in the analysis and the integration of functional connectivity informed by underlying anatomical connections. In this discussion, the authors addressed some of my concerns. For instance, while *c-Fos* immunohistochemistry and fMRI may currently be the most comparable methods available, they still differ in terms of temporal resolution.

Below are some aspects to clarify/add additional information:

Methods:

1. Human Sample: Please provide additional details regarding the inclusion and exclusion criteria for the study participants, as well as socio-demographic information about the subjects included in the analysis. Additionally, include a comparison between responders and non-responders to better understand any potential differences between these groups.

We have now added details about the inclusion and exclusion criteria for the human study:

“Inclusion criteria for participants were as follows: (i) participants must be over the age of 18, (ii) understand English and (iii) be able to provide written and verbal informed consent. The exclusion criteria were as follows: (i) not be currently or possibly pregnant, (ii) not have any metallic implants and (iii) not currently experiencing pain or have been diagnosed with any chronic pain condition.”

The only socio-demographic information collected from the participants was age and self-reported sex. Neither of these factors contributed responsiveness to either placebo or nocebo. We have now included the following in the methods section: “From our sample, 22/46 participants were identified as placebo responders (46%), and 14/25 participants identified as nocebo responders (56%). Neither age nor sex significantly influenced the likelihood of being classified as a responder in either condition (see Supplementary Table 1)” and included the following table:

Supplementary Table 1: Demographic data and measures of pain expectation for each cream prior to fMRI scans. Two-way, repeated measures ANOVAs with post-hoc multiple comparisons (Tukey correction) were used to compare expectations between creams, and between responders and non-responders. Unpaired t-tests were used to compare the age between responders and non-responders, and Fisher’s exact test was used to compare distributions of sexes between responders and non-responders.

Demographics (Placebo)	Age	Sex	
Placebo Responder (n=22)	23.95±0.71	14M / 8F	
Placebo Non-responder (n=24)	24.04±0.94	10M / 15F	
Responder vs. Non-responder	p=0.94	p=0.15	
Demographics (Nocebo)	Age	Sex	
Nocebo Responder (n=14)	22.29±0.91	8M / 6F	
Nocebo Non-responder (n=11)	23.27±1.09	4M 7F	
Responder vs. Non-responder	p=0.50	p=0.43	
Expectation (Placebo)	Vaseline	Lidocaine	Vaseline vs. Lidocaine
Placebo Responder (n=22)	49.35±0.77	33.48±1.63	p<0.0001
Placebo Non-responder (n=24)	51.67±1.76	37.14±2.71	p<0.0001
Responder vs. Non-responder	p=0.82	p=0.52	

Expectation (Nocebo)	Vaseline	Capsaicin	Vaseline vs. Capsaicin
Nocebo Responder (n=14)	48.33±0.90	68.33±2.85	p<0.0001
Nocebo Non-responder (n=11)	48.00±1.90	62.50±3.10	p=0.0008
Responder vs. Non-responder	p=0.99	p=0.30	

2. Human experiments: Regarding the timing of the sessions, were all sessions spaced one day apart? Please clarify this and provide more details, including the average interval between session 1 and session 2.

The two response-conditioning sessions were all spaced 1 day (24hr) apart for all participants. This has now been clarified and more details have been added in the 'Placebo and Nocebo Conditioning' section in the methods.

3. VOI generation and beta-value extraction – The authors wrote 'A measurement of average signal intensity, that is, relative increases or decreases in neuronal activation during the application of noxious stimuli relative to baseline periods of the fMRI scan, ' - what was considered 'baseline' in this case?

"Baseline" was considered within the GLM as any period in which thermal pain was not being applied. In our case this is the 90-seconds preceding the onset of the initial stimulus, and then every 15-second interval between each pain stimulus. This has now been clarified within the manuscript methods:

"A measurement of average signal intensity, that is, relative increases or decreases in neuronal activation during the application of noxious stimuli relative to baseline periods of the fMRI scan (i.e., the 90-second period preceding the first noxious stimulus onset, and every 15-second interval between noxious stimuli), were extracted from each of the matched control and placebo or control and nocebo contrast images."

4. Placebo and nocebo test – Were the expectations/expectancies measured? How did they relate to the placebo/nocebo effect recorded?

Expectancy was measured preceding each functional scan and has been reported in previously published material (Crawford et al., 2021; Crawford et al., 2023). Despite both groups expecting a pain reduction to the lidocaine cream and pain increase for the capsaicin cream, only placebo and nocebo responders respectively demonstrated placebo and nocebo responses. We have now included the following paragraph in the results section of the manuscript, and we have added the Supplementary Table 1 (above):

"This conditioning protocol effectively altered participants' expectations, as evidenced by significant differences in expected pain ratings for both the placebo and nocebo manipulations. There were no significant differences in expectations between placebo responders and non-responders, or between nocebo responders and non-responders, indicating that expectation changes occurred across all participants regardless of their behavioral response (Supplementary Table 1)."

5. Placebo and nocebo-induced changes in regional activity – While I agree that the use of estimation statistics provides a nuanced understanding of neural activity, I wonder whether alternative approaches, such as multivariate pattern analysis (e.g., brain signatures), could offer comparable information without requiring a predefined selection of regions to focus on. Please clarify the rationale for selecting 20 regions out of the 70 studied.

We appreciate the reviewer's suggestion regarding multivariate pattern analysis. While MVPA is a valuable approach for identifying distributed activation patterns, we chose estimation statistics to facilitate direct comparisons between species, quantify effect sizes at the regional level, and maintain biological interpretability given the anatomical differences between rodent c-Fos expression and human fMRI. This approach allowed us to examine network dynamics while preserving clarity in cross-species comparisons. However, we acknowledge the potential of MVPA and will consider its application in future studies to explore distributed neural signatures of placebo and nocebo effects.

The selection of 20 regions to compare from the 70 studied was based on the approximate number of regions that show consistent involvement in placebo and nocebo effects from previous studies and meta-analyses. We have added the following to the methods:

"We chose to select specifically 20 out of the 70 regions investigated as this is roughly the number of regions across the brain that have shown consistent involvement in placebo and nocebo effect based on previous key studies and meta-analyses (Tinnermann et al., 2017; Xu et al., 2018; Zeng et al., 2018; Crawford et al., 2021; Zunhammer et al., 2021; Crawford et al., 2023a)."

6. Please clarify the definition of the term "control group" as used throughout the text

The following clarification has been added to the manuscript:

"For humans, the 'control' therefore refers to the fMRI scan in which this moderate pain intensity was delivered to the Vaseline cream site."

"For rats, the control group thus denotes the rats that were tested during both the conditioning procedure and on Test Day at the moderate intensity stimulus."

Discussion:

1. Please expand on the potential bias introduced by using rats with chronic constriction injury of the sciatic nerve compared to healthy human participants. How might this difference affect the generalizability of the findings?

We have added the following paragraph to the discussion:

"One notable difference between the human and rodent models is that the rats were given a CCI while the human participants were healthy and uninjured. Importantly, for our purposes, both human and rodent paradigms used in our study relied on acute stimulus-evoked responses rather than spontaneous pain, making them comparable in assessing placebo- and nocebo- induced pain modulation. However, it remains possible that the CCI model alters neural processing in response to pain, placebo, and nocebo, which could introduce differences in activation patterns compared to healthy individuals. While there has yet to be an animal study that investigates whether this is the case, (Wang et al., 2022) demonstrated that participants with chronic neuropathic pain (TMD) exhibit comparable levels of placebo analgesia to healthy individuals, indicating that the neural circuitry underlying acute placebo responses remains intact in people despite the presence of chronic pain. This suggests that while chronic pain conditions may influence certain aspects of pain processing, they do not necessarily impair the fundamental mechanisms driving placebo effects. Given these considerations, it would be worthwhile to replicate our findings using uninjured rodent models, particularly since recent studies have developed more directly comparable models of placebo analgesia and nocebo hyperalgesia in healthy mice {Kimmey, 2025 #79; Poulson, 2025 #80}. Expanding research to include both injured and uninjured models will help clarify the extent to which chronic pain state influences placebo and nocebo mechanisms and enhance the generalizability of these findings."

2. One of the key questions in the field of placebo effect research is the exploration of interindividual differences. While this study did not aim to address this issue, it would be valuable to consider how the findings might contribute to our understanding of these variations in future research.

We agree with the review about the intrinsic value of considering the inter-individual differences. However, given our analysis which relies of group-level statistical models it is beyond the scope of this study and would likely detract from the key themes of our discussion, already constrained by the word limit.

3. Additionally, while the impact of verbal suggestions cannot be directly studied, the interactions between both humans and rats with the experimenter may influence the results. Was this potential confounding factor accounted for in the study design? If so, please provide details on how it was managed.

This potential confounding factor was accounted for in the study design by having the same experimenter perform all rodent work (DB) and the same experimenter perform all human procedures (LC). We have now included the following sentences in the methods to address this:

"To control for potential experimenter-induced effects, the same experimenter (DB) conducted all handling, husbandry, and procedures across all rodent groups, ensuring consistency in interactions and minimizing variability in conditioning and testing."

"Additionally, given that it has previously been shown that participant-experimenter interactions (including gender) can influence placebo and nocebo responses, the same experimenter (LC) and radiographer carried out all procedures for all participants."

Yours sincerely,

Damien Boorman, PhD
School of Medical Sciences (Neuroscience)

Damien Boorman, PhD
School of Medical Sciences (Neuroscience)
The Brain and Mind Centre
100 Mallett Street, Camperdown, NSW, Australia, 2006

15 March 2025

Dear reviewers,

We thank you both for a timely and gratifying review process. Here are the final changes made to the manuscript.

Reviewer #1 Comments:

The author's response is comprehensive and robust, addressing the reviewer's key concerns (e.g., methodological details, statistical power, study limitations) through detailed revisions and thoughtful discussions. The manuscript's transparency and scientific quality are significantly enhanced by these revisions. Although some responses could be more concise, and the discussion of future research and alternative methods more specific, the author has thoroughly resolved the reviewer's main concerns. Overall, the response is appropriate and lays a solid foundation for the manuscript's acceptance.

The following paragraph has been added to the discussion:

Future studies could potentially help to bridge these methodological differences by using fMRI in rats. However, despite significant advancements in small animal MRI imaging techniques, including the possibility of freely behaving fMRI and PET scanning, challenges such as motion artifacts, limited spatial resolution, and the technical and financial constraints of high-field imaging remain key obstacles to widespread implementation.

Reviewer #2 Comments:

All my concerns and comments were addressed.

We are glad that we were able to address the reviewers concerns and comments.

Yours sincerely,

Damien Boorman, PhD
School of Medical Sciences (Neuroscience)

**Brain and Mind Centre
School of Medical Sciences
(Neuroscience)**

**Faculty of Medicine and Health
Rm 507, 94-100 Mallett Street, M02G
Camperdown
NSW 2050 Australia**

T +61 2 9036 5314
M +61 425 222 749
E dboo2217@uni.sydney.edu.au

ABN 15 211 513
464
CRICOS 00026A